# INSTANCE-DEPENDENT CONTINUOUS-TIME REINFORCEMENT LEARNING VIA MAXIMUM LIKELIHOOD ESTIMATION

## ABSTRACT

Continuous-time reinforcement learning (CTRL) provides a natural framework for sequential decision-making in dynamic environments where interactions evolve continuously over time. While CTRL has shown growing empirical success, its ability to adapt to varying levels of problem difficulty remains poorly understood. In this work, we investigate the instance-dependent behavior of CTRL and introduce a simple, model-based algorithm built on maximum likelihood estimation (MLE) with a general function approximator. Unlike existing approaches that estimate system dynamics directly, our method estimates the state marginal density to guide learning. We establish instance-dependent performance guarantees by deriving a regret bound that scales with the total reward variance and measurement resolution. Notably, the regret becomes independent of the specific measurement strategy when the observation frequency adapts appropriately to the problem's complexity. To further improve performance, our algorithm incorporates a randomized measurement schedule that enhances sample efficiency without increasing measurement cost. These results highlight a new direction for designing CTRL algorithms that automatically adjust their learning behavior based on the underlying difficulty of the environment.

## 1 INTRODUCTION

Many real-world systems—such as autonomous robots, financial markets, and medical interventions—evolve in continuous time, where actions and feedback unfold without discrete intervals. This motivates the study of continuous-time reinforcement learning (CTRL), a framework where the agent learns to interact with a dynamic environment in real time to maximize cumulative reward. Unlike its discrete-time counterpart, CTRL is grounded in the natural temporal structure of many applications, making it particularly well-suited for control in physical and continuous systems. Recent work has highlighted its empirical potential, drawing on tools from continuous control theory (Greydanus et al., 2019; Yildiz et al., 2021; Lutter et al., 2021; Treven et al., 2024a) and the emerging use of diffusion-based models (Yoon et al., 2024; Xie et al., 2023). These developments underscore CTRL's growing relevance and its advantage in capturing fine-grained interactions that discrete-time methods often approximate only coarsely.

In this paper, we focus on the adaptivity of CTRL—that is, the ability of a learning algorithm to adjust its behavior and complexity in response to the difficulty of the problem instance. Intuitively, simpler environments should require less exploration and faster convergence, while more complex dynamics or reward structures may demand prolonged learning and finer control. For example, in robotic manipulation, navigating an open space may require significantly less precision and feedback sensitivity compared to threading a needle or interacting with deformable objects. Despite its importance, adaptivity remains largely underexplored in the CTRL literature: existing methods often lack theoretical guarantees or empirical mechanisms to modulate learning effort according to task complexity. This motivates our first core question:

> *Can we design a CTRL algorithm that is provably adaptive to problem difficulty, offering instance-dependent performance guarantees?*

A natural starting point to investigate adaptivity in CTRL is to approximate the continuous-time process using discrete-time reinforcement learning with equidistant observations. This enables us to

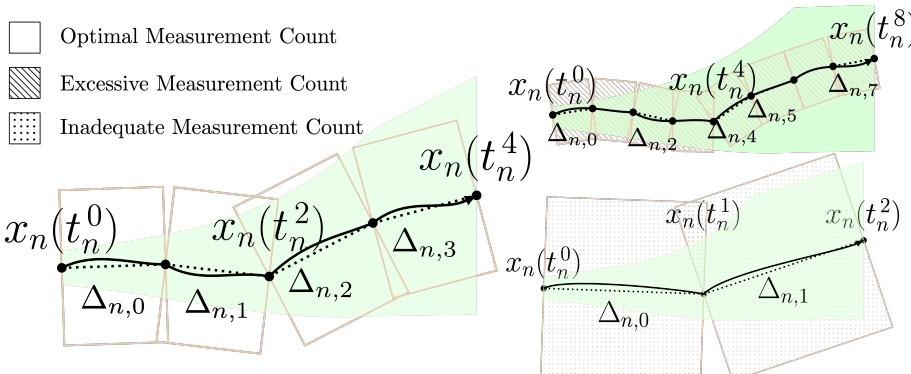

Figure 1: We depict the state trajectory $x_n(t)$ over $t \in [0, T]$ in episode $n$, with $x_n(0) = x_{\text{ini}}$ and $x_n(T)$ at the endpoints. Observation times $t_n^k$ are marked by black dots. Each measurement interval $\Delta_{n,k} = t_n^{k+1} - t_n^k$ is overlaid by a brown rectangle of width $\Delta_{n,k}$ and height proportional to $\Delta_{n,k}$, so that its area encodes $\Delta_{n,k}^2$ in our regret bound. The green shading illustrates the total variance $\text{Var}^{u_n}$. Proper measurement gap should be selected in accordance with policy variance $\text{Var}^{u_n}$ to achieve an optimal instance-dependent performance.

draw on the extensive literature on adaptivity in discrete-time RL, where regret bounds and learning dynamics have been thoroughly analyzed (Zhao et al., 2023; Zhou et al., 2023; Wang et al., 2024b;a). However, existing CTRL formulations typically apply a fixed, uniform measurement scheme to all environments, ignoring the heterogeneity in their underlying dynamics. For systems with unevenly evolving trajectories, fixed-interval sampling may either miss important events or expend effort on redundant measurements. This lack of adaptivity prevents CTRL methods from tailoring their measurement schedule to the actual variability of the environment. Consequently, a key question arises:

*How does the choice of measurement strategy in CTRL influence its ability to adapt across problem instances?*

In this work, we aim to address the two core questions outlined above. Our main contributions are summarized as follows.

- We introduce a conceptually simple model-based algorithm for CTRL, termed CT-MLE (Continuous-Time Reinforcement Learning with Maximum Likelihood Estimation). Unlike previous methods that estimate the underlying system dynamics directly (Treven et al., 2024a; Zhao et al., 2025), CT-MLE instead estimates the marginal state density using maximum likelihood estimation (MLE) with a general function approximator (e.g., neural networks or kernel models). This shift—from modeling dynamics to modeling marginal distributions—offers greater modeling flexibility and improved sample efficiency in practice. Additionally, CT-MLE is modular and compatible with a broad range of policy classes and sampling strategies, making it applicable to a wide variety of CTRL settings.

- From a theoretical perspective, we establish a regret bound for CT-MLE over the first $N$ episodes of interaction. Specifically, we show that the regret satisfies

$$\widetilde{O}\left(d^2 + d\sqrt{\sum_{n=1}^{N}\sum_{k=0}^{m_n-1}\Delta_{n,k}^2 + \sum_{n=1}^{N}\text{Var}^{u_n}}\right),$$

where $d$ denotes the complexity of the function class used for marginal density estimation, $m_n$ represents the number of measurements in episode $n$, $\Delta_{n,k}$ represents the $k$-th measurement gap in episode $n$, and $\text{Var}^{u_n}$ quantifies the total variance of the integrated reward under policy $u_n$. A central insight of our analysis is that when the measurement schedule is adapted to the problem instance—i.e., when $\sum_{k=0}^{m_n-1}\Delta_{n,k}^2$ is chosen in accordance with $\text{Var}^{u_n}$—the regret becomes primarily dependent on the reward variance and is nearly independent of the measurement schedule itself. This instance-dependent property highlights a key distinction from traditional discrete-time reinforcement learning, where measurements are typically uniform and agnostic to problem complexity. Figure 1 provides a demonstration of this phenomenon. Our results underscore the

importance of adaptive measurement strategies for achieving instance-optimal performance in the continuous-time setting.

- A core technical innovation in CT-MLE is its Monte Carlo-type randomized measurement strategy, which augments the default measurement grid with additional observation points sampled within each interval. This randomization enables unbiased estimation of the reward integral across each measurement gap, while maintaining the total number of measurements (i.e., measurement complexity) at the same order. This design not only enhances the practical effectiveness of CT-MLE but also introduces a general technique that may be of independent interest for continuous-time decision-making problems.

**Notation.** We use lower case letters to denote scalars, and use lower and upper case bold face letters to denote vectors and matrices respectively. We denote by $[n]$ the set $\{1, \ldots, n\}$. For two positive functions $a(x)$ and $b(x)$ defined on a common domain, we write $a(x) \lesssim b(x)$ if there exists an absolute constant $C > 0$ such that $a(x) \leq Cb(x)$ for all $x$ in the domain. Given a distribution $p(x)$, we use $\mathbb{E}_{x \sim p}[\cdot]$ to denote expectation and $\mathbb{V}_{x \sim p}[\cdot]$ to denote variance. For two distributions $p$ and $q$, we define their squared Hellinger distance as $\mathbb{H}^2(p \,\|\, q) := 1 - \int \sqrt{p(x)q(x)}\, dx$.

## 2 ADDITIONAL RELATED WORK

### 2.1 CONTINUOUS-TIME REINFORCEMENT LEARNING

Our work resides within the paradigm of CTRL, a foundational research thread in the control community. Early studies emphasized planning in analytically tractable settings such as the linear–quadratic regulator (LQR) (Doya, 2000; Vrabie & Lewis, 2009; Faradonbeh & Faradonbeh, 2023; Caines & Levanony, 2019; Huang et al., 2024; Basei et al., 2022; Szpruch et al., 2024). A pivotal advance occurred when Chen et al. (2018) introduced neural function approximation for learning nonlinear dynamics and value functions, thereby catalysing data-driven CTRL. Building on this foundation, Yildiz et al. (2021) proposed an episodic model-based framework that alternates between fitting ODE models to collected trajectories and solving the resulting optimal-control problem with a continuous-time actor–critic. Subsequently, Holt et al. (2024) showed that under costly observations, uniform time sampling is suboptimal and that state-dependent schedules can yield higher returns. Parallel efforts (Karimi, 2023; Ni & Jang, 2022; Holt et al., 2023) have bridged continuous-time theory with practical implementations by considering deterministic systems with discrete measurements or control updates. More recent analyses have extended these ideas to deterministic and stochastic dynamics with nonlinear approximation (Treven et al., 2024a;b), and Zhao et al. (2025) further broadened the approximation class while relaxing assumptions on epistemic-uncertainty estimators. Yet the existing theory largely provides worst-case guarantees. We close this gap by establishing the first variance-aware, nearly horizon-free *second-order* regret bound for stochastic CTRL under general function approximation—measured via the eluder dimension—and show that a simple, standard MLE-based model-based algorithm attains this bound.

### 2.2 VARIANCE-AWARE REINFORCEMENT LEARNING

There has been a series of work studying variance-aware or horizon-free sample complexity for discrete-time reinforcement learning (Simchowitz & Jamieson, 2019; Jin et al., 2020; Dann et al., 2021; Xu et al., 2021; Wagenmaker et al., 2022; He et al., 2021a;b; Zhou et al., 2021b;a; Zhao et al., 2022; Zhou & Gu, 2022). To mention a few, early online-learning work provided second-order bounds: Cesa-Bianchi et al. (2007) derived refined regret bounds based on squared losses in expert advice, and Ito et al. (2020) established tight first- and second-order regret for adversarial linear bandits using Bernstein-type concentration. Extending to MDPs, Zanette & Brunskill (2019)'s EULER algorithm achieves regret scaling with the maximum per-step return variance rather than $H$, and Foster & Krishnamurthy (2021) used triangular-discrimination bonuses to obtain small-loss bounds in contextual bandits. For structured function approximation, Kim et al. (2022) obtained horizon-free, variance-adaptive regret for linear mixture MDPs via weighted least-squares, Zhao et al. (2023) provided computationally efficient variance-dependent bounds for linear bandits and mixtures, and Zhang et al. (2021) devised variance-aware confidence sets giving logarithmic horizon dependence. Distributional RL has delivered second-order guarantees under general classes by modeling full return distributions (Zhang et al., 2022), and Huang et al. (2024) achieved sublinear regret for continuous-time stochastic LQR by estimating transition. Despite these advances, all require specialized variance or distributional machinery; our work shows that a standard MLE-based

model-based RL approach attains nearly horizon-free, second-order variance-dependent bounds under general function approximation without bespoke variance estimation or distributional techniques, similar to Wang et al. (2024b) but under the continuous-time setup.

## 3 PROBLEM SETUP

**Stochastic Differential Equation Formulation.** We consider a general nonlinear continuous-time dynamical system governed by a stochastic differential equation (SDE). Let $x(\cdot)$ denote the state trajectory over a fixed planning horizon $[0, T]$, where $x(t) \in \mathcal{X} \subseteq \mathbb{R}^l$ for all $t \in [0, T]$. The system dynamics under a deterministic policy $u \in \Pi : \mathcal{X} \to \mathcal{U} \subseteq \mathbb{R}^r$ are described by

$$dx(t) = f(x(t), u(x(t))) \, dt + g(x(t), u(x(t))) \, dw(t),$$

where $w(t) \in \mathbb{R}^l$ is a standard Wiener process and the SDE is interpreted in the Itô sense. Here, $f \in \mathcal{F}$ and $g \in \mathcal{G}$, where each $f : \mathcal{X} \times \mathcal{U} \to \mathbb{R}^l$ and $g : \mathcal{X} \times \mathcal{U} \to \mathbb{R}^{l \times l}$ denote the drift and diffusion functions, respectively. Given an initial state $x(0) = x$, we denote by $p_{f,g}(u, x)$ the law of the trajectory $x(\cdot)$. We write $p_{f,g}(u, x, s)$ for the marginal distribution of $x(s)$ and use $p_{f,g}(\cdot \mid u, x, s)$ to denote its corresponding density function.

**Learning Protocol.** The learning process unfolds in episodes. In each episode $n = 1, \ldots, N$, the agent executes a policy $u_n$ and observes the trajectory $x(\cdot) \sim p_{f^*, g^*}(u_n, x_{\text{ini}})$, where $(f^*, g^*)$ denotes the unknown environment and $x_{\text{ini}}$ is the fixed initial state. During execution, the agent selects a set of measurement times $\{t_n^k\}_{k=1}^{m_n} \subset [0, T]$ at which observations are collected. These observations are used to update the policy for the next episode. The agent's objective is to find a policy that maximizes the expected cumulative reward under the reward function $b : \mathcal{X} \times \mathcal{U} \to \mathbb{R}$:

$$u^* = \arg\max_{u \in \Pi} R_{f^*, g^*}(u), \quad \text{where} \quad R_{f,g}(u) := V_{f,g}(u, x_{\text{ini}}, 0),$$

and the value function is given by

$$V_{f,g}(u, x, s) := \mathbb{E}_{x(\cdot) \sim p_{f,g}(u, x_{\text{ini}})} \left[ \int_{t=s}^{T} b(x(t), u(x(t))) \, dt \,\middle|\, x(s) = x \right].$$

**Performance Metrics.** We evaluate algorithmic performance using several metrics. *The regret* is defined as

$$\text{Regret}(N) := \sum_{n=1}^{N} \left( R_{f^*, g^*}(u^*) - R_{f^*, g^*}(u_n) \right),$$

We say a policy $u$ is $\epsilon$-optimal if $R_{f^*, g^*}(u^*) - R_{f^*, g^*}(u) \leq \epsilon$. For any CTRL algorithm that returns an $\epsilon$-optimal policy after $N$ episodes, we define the *episode complexity* as $N$, and the *measurement complexity* as $\sum_{n=1}^{N} m_n$, where $m_n$ denotes the number of measurements in episode $n$. We also consider the *$\lambda$-total complexity* for any $\lambda \in [0, 1]$, defined as the weighted sum: $(1 - \lambda)N + \lambda \sum_{n=1}^{N} m_n$. This interpolates between pure episode complexity ($\lambda = 0$) and pure measurement complexity ($\lambda = 1$).

## 4 CTRL WITH MAXIMUM LIKELIHOOD ESTIMATION

In this section, we introduce our algorithm, CT-MLE, as described in Algorithm 1. At a high level, each episode $n$ follows the standard optimistic model-based approach in CTRL (Treven et al., 2024b). Specifically, the agent constructs a confidence set for the unknown drift $f^*$ and diffusion $g^*$, and then applies the principle of optimism to select a near-optimal policy $u_n \in \Pi$. Such an optimization step requires an oracle access to maximize over joint sets of policy $u$ and dynamics $f, g$, which are standard in literature (Treven et al., 2024a; Jin et al., 2021; Abbasi-Yadkori et al., 2011). The selected policy is executed in the environment, yielding a continuous-time trajectory $x_n(\cdot)$. The agent then collects informative observations from this trajectory to refine its confidence set for the next episode. This framework parallels optimistic approaches in discrete-time RL (Abbasi-Yadkori et al., 2011; Jin et al., 2019; Russo & Van Roy, 2013; Jin et al., 2021), though applied to the continuous-time setting.

A key distinction in CTRL is that the agent must decide *when* to observe the trajectory, since data is generated in continuous time. To address this, CT-MLE introduces a sequence of measurement times $(t_n^k)_{k=1}^{m_n}$ for each episode $n$. The agent collects observations only at these time points, i.e., $\{x_n(t_n^k)\}_{k=1}^{m_n}$. Importantly, we allow the measurement times to be non-uniformly spaced, meaning the measurement gap $\Delta_{n,k} := t_n^{k+1} - t_n^k$ can vary across time.

---

**Algorithm 1** Continuous-Time Reinforcement Learning with Maximum Likelihood Estimation

---

**Require:** Episode number $N$, policy class $\Pi$, initial state $x_{\text{ini}}$, drift class $\mathcal{F}$, diffusion class $\mathcal{G}$, reward function $b$, confidence radius $\beta$, planning horizon $T$.

1: For each $n \in [N]$, determine a fixed measurement time sequence $0 = t_n^0 < \cdots < t_n^{m_n} = T$. For any $0 \leq k < m_n$, denote measurement gaps $\Delta_{n,k} := t_n^{k+1} - t_n^k$.
2: **for** episode $n = 1, \ldots, N$ **do**
3:   Set confidence sets of $(f, g)$ as $\mathcal{P}_n$, where

$$
\mathcal{P}_n := \Bigg\{ (f, g) \in \mathcal{F} \times \mathcal{G} : \sum_{i=1}^{n-1} \sum_{k=0}^{m_i-1} \log p_{f,g}(x_i(t_i^{k+1}) | u_i, x_i(t_i^k), \Delta_{i,k})
$$

$$
\geq \max_{(f',g') \in \mathcal{F} \times \mathcal{G}} \sum_{i=1}^{n-1} \sum_{k=0}^{m_i-1} \log p_{f',g'}(x_i(t_i^{k+1}) | u_i, x_i(t_i^k), \Delta_{i,k}) - \beta \Bigg\}.
$$

4:   (Randomized strategy) set $\widehat{\mathcal{P}}_n$ following Algorithm 2.
5:   Set policy $u_n, f_n, g_n$ as $u_n, f_n, g_n = \arg\max_{u \in \Pi, (f,g) \in \mathcal{P}_n \cap \widehat{\mathcal{P}}_n} R_{f,g}(u)$.
6:   Execute the $n$-th episode and observe $x_n(t_n^0), \ldots, x_n(t_n^{m_n})$.
7:   (Randomized strategy) obtain additional observations to build $\widehat{\mathcal{P}}_n$ following Algorithm 2.
8: **end for**
9: **return** Randomly pick an $n \in [N]$ uniformly and output $\widehat{u}$ as $u_n$.

---

**Maximum Likelihood Estimation.** To construct the confidence set, we begin by examining the learning objective in CTRL. Due to the Markov property of the Itô process, for any drift-diffusion pair $(f, g)$, policy $u$, state $x$, time $s$, and measurement gap $\Delta$, the following identity holds:

$$
V_{f,g}(u, x, s) = \mathbb{E}_{x' \sim p_{f,g}(u, x, \Delta)} \left[ V_{f,g}(u, x', s + \Delta) \right] + \mathbb{E}_{x(\cdot) \sim p_{f,g}(u, x)} \left[ \int_{t=0}^{\Delta} b(x(t), u(x(t))) \, dt \right]. \quad (4.1)
$$

A detailed derivation for equation 4.1 is provided in Appendix A.4. This can be viewed as a continuous-time analogue of the Bellman equation. It implies that to evaluate the value function $V_{f^*,g^*}(u, x_{\text{ini}}, 0)$, it suffices to estimate the marginal distribution $p_{f^*,g^*}(u, x, \Delta)$ and the trajectory distribution $p_{f^*,g^*}(u, x)$ over the interval $[0, \Delta]$. To estimate the first term in equation 4.1, we construct a confidence set $\mathcal{P}_n$ based on MLE over historical observations, as defined in line 3 of Algorithm 1, inspired by existing works about MLE for discrete-time RL (Agarwal et al., 2020; Liu et al., 2022; Wang et al., 2024a;b). Specifically, $\mathcal{P}_n$ contains all drift-diffusion pairs $(f, g)$ whose likelihood on the conditional distribution $p_{f,g}(x_i(t_i^{k+1}) \mid u_i, x_i(t_i^k), \Delta_{i,k})$ is sufficiently close to that of the MLE solution. The proximity is controlled via a confidence radius parameter $\beta$.

We note that existing approaches (Treven et al., 2024a; Zhao et al., 2025) typically aim to learn the underlying dynamics $(f^*, g^*)$ by directly estimating the drift term $f^*(x(t))$. In the corresponding deterministic setting where the diffusion term is zero, this drift is equivalent to the time derivative $\dot{x}(t)$. However, estimating this term from discrete and noisy trajectory data often requires non-trivial procedures like finite-difference approximations, which introduces additional algorithmic complexity and sensitivity to noise. In contrast, our approach relies solely on the observed states at discrete measurement times, making the estimation process both simpler and more robust.

**Randomized Additional Measurement.** The second term in equation 4.1 involves an integral over the trajectory segment $x(\cdot)$ governed by the law $p_{f,g}(u, x)$. While this integral could in principle require full knowledge of the process, it can instead be estimated using a single sample point via a Monte Carlo-style approach. To implement this, we augment CT-MLE with an additional randomized measurement step, as described in Algorithm 2. Specifically, for each interval $[t_i^k, t_i^{k+1})$, we sample a random time $\widehat{t}_{i,k} = t_i^k + \widehat{\Delta}_{i,k}$ uniformly from the interval and record the state $x_i(\widehat{t}_{i,k})$. It is worth noting that this modification requires only one additional measurement per interval, effectively doubling the number of measurements compared to CT-MLE without Algorithm 2. Using these additional samples, we construct a second confidence set $\widehat{\mathcal{P}}_n$, based on the conditional distribution $p_{f,g}(x_i(\widehat{t}_{i,k}) \mid u_i, x_i(t_i^k), \widehat{\Delta}_{i,k})$. Notably, our algorithm does not explicitly compute the integral in equation 4.1; instead, the randomized measurements serve to implicitly capture the integral's behavior by refining the confidence set around the true dynamics $(f^*, g^*)$. This enables us to eliminate the continuity assumption without compromising performance guarantees.

---

**Algorithm 2** Monte Carlo-Type Estimation

---

**Require:** Current episode $n$, history observations $\{x_i(t_i^k), x_i(t_i^k + \widehat{\Delta}_{i,k})\}_{i=1,\ldots,n-1,k=0,\ldots,m_i-1}$, measurement gaps $\{\Delta_{n,k}\}_{k=0,\ldots,m_n-1}$.

1: Build confidence set $\widehat{\mathcal{P}}_n$ as

$$\widehat{\mathcal{P}}_n := \left\{ (f,g) \in \mathcal{F} \times \mathcal{G} : \sum_{i=1}^{n-1} \sum_{k=0}^{m_i-1} \log p_{f,g}(x_i(t_i^k + \widehat{\Delta}_{i,k})|u_i, x_i(t_i^k), \widehat{\Delta}_{i,k}) \right.$$

$$\left. \geq \max_{(f',g') \in \mathcal{F} \times \mathcal{G}} \sum_{i=1}^{n-1} \sum_{k=0}^{m_i-1} \log p_{f',g'}(x_i(t_i^k + \widehat{\Delta}_{i,k})|u_i, x_i(t_i^k), \widehat{\Delta}_{i,k}) - \beta \right\}.$$

2: Set $\widehat{\Delta}_{n,k} \sim \text{Unif}(0, \Delta_{n,k})$ for all $0 \leq k < m_n$.
3: **return** Confidence set $\widehat{\mathcal{P}}_n$ and observations $x_n(\widehat{t}_n^0 + \widehat{\Delta}_{n,0}), \ldots, x_n(t_n^{m_n-1} + \widehat{\Delta}_{n,m_n-1})$.

---

## 5 ANALYSIS OF CT-MLE

We present the theoretical analysis of Algorithm 1. We begin by introducing the following regularity assumption, which summarizes all the conditions we impose on the system dynamics.

**Assumption 5.1.** *The continuous-time system dynamics satisfy the following conditions:*

- *The reward function $b(x,u)$ and the initial state $x_{ini}$ are known to the agent.*

- *The reward function is bounded: $0 \leq b(x,u) \leq 1$ for all $(x,u) \in \mathcal{X} \times \mathcal{U}$. Furthermore, for any trajectory $x(\cdot) \sim p_{f^*,g^*}(u, x_{ini})$, the cumulative reward is bounded as $\int_0^T b(x(t), u(x(t)))\, dt \leq 1$.*

**Remark 5.2.** The boundedness assumption on $b$ is made for simplicity. For any general reward function $b$ satisfying $0 \leq b(x,u) \leq B_1$ and $\int_0^T b(x(t), u(x(t)))\, dt \leq B_2$, one can normalize the reward by defining $b' := b/\max(B_1, B_2)$ and apply the algorithm and analysis to $b'$.

Next, we introduce the notion of *total variance* for a policy $u$, a concept originating from discrete-time reinforcement learning (Wang et al., 2024b; Zhou et al., 2023), which serves as an instance-dependent measure of problem hardness.

**Definition 5.3.** *For any policy $u \in \Pi$, we define its total variance $\text{Var}^u$ and the maximal total variance $\text{Var}^\Pi$ as*

$$\text{Var}^u := \mathbb{V}_{x(\cdot) \sim p_{f^*,g^*}(u, x_{ini})} \left[ \int_0^T b\big(x(t), u(x(t))\big)\, dt \right], \quad \text{Var}^\Pi := \max_{u \in \Pi} \text{Var}^u.$$

By Assumption 5.1, it immediately follows that $\text{Var}^u \leq 1$ for any $u \in \Pi$. The total variance $\text{Var}^u$ quantifies the uncertainty in the cumulative reward under the stochastic dynamics, and is tightly connected to the diffusion term $g$. The following proposition formally characterizes this dependence.

**Proposition 5.4.** *Suppose the following conditions hold:*

- *The reward function $b$ is $L_b$-Lipschitz continuous: for all $x, x' \in \mathcal{X}$ and $y, y' \in \mathcal{U}$,*

$$|b(x,y) - b(x',y')| \leq L_b \left( \|x-x'\|_2 + \|y-y'\|_2 \right).$$

- *The drift $f \in \mathcal{F}$ is $L_f$-Lipschitz continuous, and the policy $u \in \Pi$ is $L_u$-Lipschitz continuous:*

$$\|f(x,y) - f(x',y')\|_2 \leq L_f \left( \|x-x'\|_2 + \|y-y'\|_2 \right), \quad \|u(x) - u(y)\|_2 \leq L_u \|x-y\|_2.$$

- *The diffusion term $g$ has bounded Frobenius norm: $\|g(x,y)\|_F \leq G$ for all $x \in \mathcal{X}$ and $y \in \mathcal{U}$.*

*Then, for any $u \in \Pi$, the total variance is bounded as*

$$\text{Var}^u \leq \min \left\{ 1,\ G^2 \cdot \frac{TL_b^2(1+L_u)}{2L_f} \left( e^{2L_f(1+L_u)T} - 1 \right) \right\}.$$

Proposition 5.4 shows that the total variance $\text{Var}^u$ is controlled by the magnitude of the diffusion term $G$. In particular, in a deterministic environment ($G = 0$), we have $\text{Var}^u = 0$ for all $u \in \Pi$. Furthermore, if the policy $u$ is less sensitive to its input (i.e., has small $L_u$), the total variance is also

reduced. These observations support the use of $\mathrm{Var}^u$ as a meaningful measure of instance difficulty in continuous-time reinforcement learning.

Next, we recall the notion of the *eluder dimension* (Russo & Van Roy, 2013; Wang et al., 2023; 2024b; Zhao et al., 2025), which we use to characterize the complexity of the system dynamics class $\mathcal{F} \times \mathcal{G}$. In addition to the eluder dimension, we also quantify the richness of the dynamics class through its *bracketing numbers* (Geer, 2000), defined as follows.

**Definition 5.5.** *Let $\Psi$ be a class of real-valued functions defined on a domain $\mathcal{Y}$. The $\epsilon$-eluder dimension $\mathrm{DE}_p(\Psi, \mathcal{Y}, \epsilon)$ is the length of the longest sequence $y^1, \ldots, y^L \subseteq \mathcal{Y}$ such that for all $t \in [L]$, there exists $\psi \in \Psi$ satisfying $\sum_{\ell=1}^{t-1} |\psi(y^\ell)|^p \leq \epsilon^p$ and $|\psi(y^t)| > \epsilon$.*

*In this work, we specify $\mathcal{Y} = \Pi \times \mathcal{X} \times [0, T]$ and define the function class $\Psi = \{\psi_{f,g}\}_{(f,g) \in \mathcal{F} \times \mathcal{G}}$, where*

$$\psi_{f,g}(u, x, t) := \mathbb{H}^2\left(p_{f,g}(u, x, t) \,\|\, p_{f^*, g^*}(u, x, t)\right).$$

*For notational convenience, we write $d_{1/\epsilon}$ to denote $\mathrm{DE}_1(\Psi, \mathcal{Y}, \epsilon)$.*

**Definition 5.6.** *Let $\Upsilon$ be a class of real-valued functions defined on the domain $\mathcal{Y} \times \mathcal{X}$. For any functions $l_1, l_2 : \mathcal{Y} \times \mathcal{X} \to \mathbb{R}$ satisfying $l_1(y, x) \leq l_2(y, x)$ for all $(y, x) \in \mathcal{Y} \times \mathcal{X}$, the bracket $[l_1, l_2] = \{v \in \Upsilon : l_1(y, x) \leq v(y, x) \leq l_2(y, x), \forall (y, x) \in \mathcal{Y} \times \mathcal{X}\}$. Given a norm $\|\cdot\|$ on functions over $\mathcal{Y} \times \mathcal{X}$, the bracket $[l_1, l_2]$ is an $\epsilon$-bracket if $\|l_2 - l_1\| \leq \epsilon$. The $\epsilon$-bracketing number of $\Upsilon$ with respect to $\|\cdot\|$, denoted $\mathcal{N}_{[]}(\epsilon, \Upsilon, \|\cdot\|)$, is the minimal number of $\epsilon$-brackets required to cover $\Upsilon$.*

*In this work, we take $\mathcal{Y} = \Pi \times \mathcal{X} \times [0, T]$ and consider the function class $\Upsilon = \{v_{f,g}\}_{(f,g) \in \mathcal{F} \times \mathcal{G}}$ with the norm $\|\cdot\|$ defined by*

$$v_{f,g}(u, x, t, x') := p_{f,g}(x' \mid u, x, t), \qquad \|v\| = \sup_{(u,x,t) \in \mathcal{Y}} \int_{x'} |v(u, x, t, x')| dx'. \tag{5.1}$$

*For notational convenience, we write $\mathcal{C}_{1/\epsilon}$ to denote $\mathcal{N}_{[]}(\epsilon, \Upsilon, \|\cdot\|)$.*

**Remark 5.7.** The function class $\Psi$ is chosen for analytical clarity. First, by assuming a known reward function (Assumption 5.1), we isolate the core challenge to learning the unknown dynamics $(f^*, g^*)$. This allows for a focused analysis of how the measurement strategy and stochasticity affect regret. While a unified analysis incorporating the reward function is common in other settings (Jin et al., 2021; He et al., 2021b), its extension to continuous time is a nontrivial challenge deferred to future work. Second, using the squared Hellinger distance provides a direct analytical bridge between the statistical error of our estimator and the regret decomposition, which is central to the proof for the final regret bound.

**Remark 5.8.** Treven et al. (2024a) introduced a model complexity notion $\mathcal{I}_N$ based on an external estimator for the epistemic uncertainty of $f^*, g^*$. In contrast, our eluder dimension requires no such estimator, offering a broader, self-contained characterization. Zhao et al. (2025) also considered eluder dimension in CTRL, but theirs targets only the nonlinearity in estimating $f^*$, while ours captures the nonlinearity of the full induced distribution $p_{f^*, g^*}$, yielding a more general measure.

We show that several natural classes of $(f, g)$ admit a small eluder dimension $d_{1/\epsilon}$ and bracketing number $\mathcal{C}_{1/\epsilon}$.

**Proposition 5.9.** *Suppose the marginal density admits the quadratic form*

$$p_{f,g}(x' \mid u, x, t) = \left(\phi(u, x, t)^\top \mu_{f,g}(x')\right)^2, \qquad \phi, \mu_{f,g} \in \mathbb{R}^d,$$

*and assume $\|\phi(u, x, t)\|_2 \leq 1$ and $\int_{x'} \|\mu_{f,g}(x')\|_2^2 \, dx' \leq B$. Then the corresponding $\psi_{f,g}$ and $v_{f,g}$ satisfy*

$$d_{1/\epsilon} \lesssim d^2 \log\left(1 + \frac{B^2}{\epsilon^2}\right), \qquad \mathcal{C}_{1/\epsilon} = |\mathcal{F}||\mathcal{G}|.$$

**Proposition 5.10.** *Suppose the marginal density admits the quadratic representation*

$$p_{f,g}(x' \mid u, x, t) = \left(\phi(u, x, t)^\top M_{f,g} \mu(x')\right)^2, \qquad \phi, \mu \in \mathbb{R}^d, M_{f,g} \in \mathbb{R}^{d \times d}.$$

*Assume $\|\phi(u, x, t)\|_2 \leq 1$, $\|\mu(x')\|_2 \leq \sqrt{B}$, $\|M_{f,g}\|_F \leq \sqrt{B}$, each coordinate of $\phi$ and $\mu$ is nonnegative, and the normalization $\int_{x'} [\mu(x')]_i \, dx' = 1$ holds for all coordinates. Then the corresponding $\psi_{f,g}$ and $v_{f,g}$ satisfy*

$$d_{1/\epsilon} \lesssim d^2 \log\left(1 + \frac{B^2}{\epsilon^2}\right), \qquad \mathcal{C}_{1/\epsilon} \lesssim \left(\frac{3d^2 B^3}{\epsilon}\right)^{d \times d}.$$

We now present our main theory.

**Theorem 5.11.** *For any fixed grid* $(t_n^k)$, *define* $\boldsymbol{\Delta}_n := \sqrt{\sum_{k=0}^{m_n-1} \Delta_{n,k}^2}$ *and* $\boldsymbol{m}_N := \sum_{n=1}^{N} m_n$. *Given* $0 < \delta < 1$, *set* $\iota := \log(N/\delta)\log(\boldsymbol{m}_N)$, $\mathcal{C}_{3\boldsymbol{m}_N} := \mathcal{N}_{[]}(1/(3\boldsymbol{m}_N), \Psi, \|\cdot\|)$ *following Definition 5.6. Then denote* $\beta = 5\log(N\mathcal{C}_{3\boldsymbol{m}_N}/\delta)$, $d_{\boldsymbol{m}_N} := \mathrm{DE}_1(\Psi, \mathcal{Y}, 1/\boldsymbol{m}_N)$ *and* $d_{8\beta\boldsymbol{m}_N} := \mathrm{DE}_1(\Psi, \mathcal{Y}, 1/(8\beta\boldsymbol{m}_N))$ *following Definition 5.5, under Assumption 5.1, with probability at least* $1 - 8\delta$, *we have*

$$Regret(N) \lesssim \iota\bigg(d_{8\beta\boldsymbol{m}_N}\beta + \sqrt{d_{\boldsymbol{m}_N}\beta\bigg(\sum_{n=1}^{N}\boldsymbol{\Delta}_n^2 + \sum_{n=1}^{N}\mathrm{Var}^{u_n}\bigg)}\bigg). \tag{5.2}$$

*Proof sketch.* We summarize the main challenges and ideas behind the proof of Theorem 5.11.

- The first challenge is the decomposition of Regret($N$), since the value function $V_{f,g}(u, x, t)$ is defined in continuous time and thus lacks the natural step-wise structure of discrete-time MDPs. We rely on the continuous-time one-step identity in equation 4.1: by the Markov property, the future trajectory depends on the past only through the current state, so the distribution of $x(s + \Delta)$ is fully characterized by the transition density $p_{f,g}(u, x, \Delta)$. Applying this recursion on the measurement grid $\{t_n^k\}_{k=0}^{m_n}$ yields a discrete sequence of one-step relations, allowing the suboptimality gap $V_{f^*,g^*}(u^*, x_{\mathrm{ini}}, 0) - V_{f^*,g^*}(u_n, x_{\mathrm{ini}}, 0)$ to be decomposed into value gaps $V_{f_n,g_n}(u_n, x_n(t_n^k), t_n^k) - V_{f_n,g_n}(u_n, x_n(t_n^{k+1}), t_n^{k+1})$ and reward-integral gaps $\mathbb{E}\big[\int_0^{\Delta_{n,k}} b(x(t), u_n(t))\, dt\big] - \int_{t_n^k}^{t_n^{k+1}} b(x_n(t), u_n(t))\, dt$.

- The value gaps can be controlled using standard techniques from discrete-time analyses. The reward-integral gaps, however, are new in continuous time. Bounding the integral $\mathbb{E}\big[\int_0^{\Delta_{n,k}} b(x(t), u_n(t))\, dt\big] - \int_{t_n^k}^{t_n^{k+1}} b(x_n(t), u_n(t))\, dt$ requires knowledge of the trajectory inside each interval, which in principle demands pointwise estimation of $p_{f^*,g^*}$. Since pointwise convergence is unattainable under typical learning guarantees, a direct approach is infeasible. To overcome this issue, Algorithm 2 augments each interval with a single auxiliary observation sampled uniformly at time $\widehat{\Delta}_{n,k}$. This randomization produces an unbiased Monte Carlo estimate of the reward integral and enables the construction of an additional likelihood-based confidence set that captures intra-interval behavior while keeping the measurement cost essentially unchanged.

- The final step combines these estimates within a regret analysis that incorporates the variance term $\mathrm{Var}^{u_n}$, which captures diffusion-driven fluctuations of the reward integral. These fluctuations accumulate at order $\Delta_{n,k}^2$, leading to the additional term $\sum_{n=1}^{N}\boldsymbol{\Delta}_n^2$ in the final regret bound. This term is intrinsic to the continuous-time dynamics and has no analogue in the discrete-time setting.

$\square$

To the best of our knowledge, the resulting regret bound of Algorithm 1 is the first *instance-dependent second-order regret bound* established in CTRL. Notably, the dependence on $\mathrm{Var}^{u_n}$ is independent of the measurement strategy, highlighting it as a fundamental quantity characterizing the intrinsic difficulty of the continuous-time system dynamics. We summarize several key remarks below.

**Remark 5.12.** The regret bound equation 5.2 remains unchanged as long as the total measurement budget $\boldsymbol{\Delta}_n$ is fixed. This implies that CTRL is *robust* to different choices of measurement schedules, provided the total measurement effort remains the same. This aligns with recent observations (Treven et al., 2024b) suggesting that CTRL is relatively insensitive to the minimum measurement gap $\min_k \Delta_{n,k}$. In particular, while equidistant measurements may seem natural—as they mirror discrete-time RL—they are not the only strategy capable of achieving near-optimal regret guarantees.

**Remark 5.13.** Many prior works on CTRL derive regret or sample complexity bounds that scale exponentially with the planning horizon $T$, i.e., contain terms of the form $\exp(T)$ (Treven et al., 2024a; Zhao et al., 2025), making the bounds vacuous for large $T$. In contrast, our regret bound in equation 5.2 depends on $T$ only *logarithmically*, due to the use of the total variance $\mathrm{Var}^{u_n}$, which is bounded by 1 under Assumption 5.1. We emphasize that avoiding the exponential dependence on $T$ is made possible by analyzing the problem through the lens of total variance. Without this perspective, one would recover an exponential dependence on $T$, as shown in Proposition 5.4.

Next we discuss a more refined version of regret bound and $\lambda$-total complexity of CT-MLE.

**Corollary 5.14.** *Using the notations defined in Theorem 5.11, suppose there exists a constant $d > 0$ such that $d \geq \max\{d_{8\beta m_N}, d_{m_N}, \beta\}$. Then selecting equidistant measurements $\Delta_{n,k} = \Delta$, the regret is bounded as*

$$Regret(N) \lesssim \log(N/\delta)\log(TN/\Delta)\left(d^2 + d\sqrt{NT\Delta + N\mathrm{Var}^\Pi}\right).$$

*Furthermore, to find an $\epsilon$-optimal policy $\widehat{u}$, the $\lambda$-total complexity is bounded, up to logarithmic factors, by*

$$(1-\lambda)\left(\frac{d^2}{\epsilon} + \frac{d^2\mathrm{Var}^\Pi}{\epsilon^2}\right) + \lambda\frac{d^2T^2}{\epsilon^2} + \frac{(1-\lambda)d^2T\Delta}{\epsilon^2} + \left(\frac{d^2}{\epsilon} + \frac{d^2\mathrm{Var}^\Pi}{\epsilon^2}\right)\frac{\lambda T}{\Delta}. \tag{5.3}$$

We have the following remarks about the total complexity equation 5.3.

**Remark 5.15.** When $\lambda = 0$, i.e., we only care about the episode complexity and ignore the measurement complexity, selecting the measurement gap as $\Delta = \mathrm{Var}^\Pi/T$ yields an episode complexity of $d^2\mathrm{Var}^\Pi/\epsilon^2$. This result suggests that to fully exploit the instance-dependent property of Algorithm 1, it suffices to choose an instance-dependent measurement gap. In particular, achieving instance-adaptive performance requires measuring more frequently in less stochastic environments. Meanwhile, the measurement complexity becomes $d^2T^2/\epsilon^2$, which is independent of the specific problem instance.

**Remark 5.16.** When $\lambda = 1$, i.e., we focus solely on the measurement complexity and ignore the episode complexity, the optimal choice is $\Delta = T$. The total measurement complexity is proportional to $\frac{d^2\mathrm{Var}^\Pi}{\epsilon^2} \cdot \frac{T}{\Delta}$. To minimize this expression, $\Delta$ must be maximized. This implies a sparse sampling strategy where for each episode, we collect samples at the start and end points, $x(0)$ and $x(T)$, along with one additional sample at a random time $\widehat{t} \in [0, T]$. This result highlights a theoretical trade-off, favoring many "measurement-cheap" episodes over a few "measurement-expensive" ones. Interestingly, the measurement complexity asymptotically matches the complexity when episode complexity is the sole focus $\lambda = 0$. This observation leads to an interesting conjecture: the problem instance influences only the episode complexity, but not the measurement complexity. Verifying the tightness of these bounds remains an open direction for future work.

## 6 CONCLUSION AND LIMITATIONS

**Conclusion.** In this work, we presented CT-MLE, a simple and general model-based algorithm for CTRL that learns through marginal density estimation rather than explicit dynamic modeling. Our approach leverages MLE with flexible function approximators, enabling compatibility with a wide range of policy classes and continuous-time settings. We introduced a randomized measurement strategy, including a Monte Carlo-style scheme that provides unbiased integral estimation while preserving measurement efficiency. Theoretically, we established regret bounds that reveal the benefit of instance-dependent measurement schedules, and we demonstrated that the regret can be made primarily dependent on total reward variance, effectively decoupling it from fixed measurement grids.

**Limitations.** While our work provides a theoretical foundation, several gaps remain. First, we assume access to general function approximators, but do not provide a computationally efficient, provably correct algorithm. A key next step is to develop an adaptive method that estimates variance online and sets measurement gaps accordingly. Second, our analysis relies on a simplified continuous-time structure for tractability, which may not hold in practice. Future work could identify realistic dynamics that still support Eluder-dimension-based analysis. Third, our framework assumes a known deterministic reward and stationary policy. Extending to stochastic rewards and time-varying policies $u(t, x)$ would require generalizing existing tools to the joint state-time domain.

## ETHICS STATEMENT

Our study develops and analyzes algorithms for continuous-time reinforcement learning (CTRL) using a theoretical SDE-based formulation and episodic learning protocol; it does not involve human subjects, personally identifiable information, or sensitive data, and all experiments are performed in simulator settings (standard RL environments) rather than on physical systems. The work focuses on algorithmic methods (Algorithm 1, 2) and formal analysis, not deployment, thereby avoiding direct safety risks in real-world control; nevertheless, we caution that applying any learned policy to safety-critical domains (e.g., robotics, healthcare, finance) should include appropriate risk assessment, domain-specific safeguards, and compliance checks.

## REPRODUCIBILITY STATEMENT

We facilitate reproducibility by referencing precise locations of all necessary components: the formal problem setup (Section 3) and learning protocol, the complete algorithmic specification (Algorithm 1 and randomized measurement Algorithm 2), and full theoretical details, assumptions, and proofs in the appendix (Appendix B with supporting lemmas). Experimental settings, implementation specifics, and environment configurations are documented in the "Numerical Experiments" appendix (Appendix C), including "Implementation Details," main results, and ablations, with further clarifications in "Additional Details". Together, these materials specify objectives, schedules, and measurement strategies sufficient to reproduce the reported results or re-create them under equivalent simulator conditions.

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

CONTENTS OF THE APPENDIX

## THE USE OF LARGE LANGUAGE MODELS (LLMs)

LLMs were used solely for language polishing; all ideas, analyses, and conclusions are the authors' own, and the authors take full responsibility for the final text.

# A    ADDITIONAL RESULTS FROM MAIN PAPER

## A.1    PROOF OF PROPOSITION 5.4

*Proof.* For any deterministic policy $u$, we have

$$
\text{Var}^u = \mathbb{E}_{x(\cdot)\sim p_{f^*,g^*}(u,x_{\text{ini}})} \left[ \int_0^T b(x(t),u(x(t))) - \mathbb{E}_{x(\cdot)\sim p_{f^*,g^*}(u,x_{\text{ini}})} \int_0^T b(x(t),u(x(t))) \right]^2.
$$

Applying the Cauchy–Schwarz inequality yields

$$
\text{Var}^u \leq \mathbb{E}_{x(\cdot)\sim p_{f^*,g^*}(u,x_{\text{ini}})} \left[ \int_0^T \left( b(x(t),u(x(t))) - \mathbb{E}_{x\sim p_t}b(x,u(x)) \right)^2 dt \right]
$$

$$
= \int_0^T \mathbb{E}_{x\sim p_t} \left( b(x,u(x)) - \mathbb{E}_{x'\sim p_t}b(x',u(x')) \right)^2 dt, \tag{A.1}
$$

where we denote $p_t = p_{f^*,g^*}(u,x_{\text{ini}},t)$ for simplicity.

For the integrand in equation A.1, by the Lipschitz continuity of $b$ and $u$, we have

$$
\mathbb{E}_{x\sim p_t} \left( b(x,u(x)) - \mathbb{E}_{x'\sim p_t}b(x',u(x')) \right)^2
$$

$$
= \mathbb{E}_{x,x'\sim p_t} \left( b(x,u(x)) - b(x',u(x')) \right)^2
$$

$$
\leq \mathbb{E}_{x,x'\sim p_t} \left( L_b(\|x-x'\|_2 + L_u\|x-x'\|_2) \right)^2
$$

$$
= L_b^2(1+L_u)^2 \mathbb{E}_{x,x'\sim p_t}\|x-x'\|_2^2
$$

$$
= 2L_b^2(1+L_u)^2 \, \mathbb{E}_{x\sim p_t} \left\| x - \mathbb{E}_{x'\sim p_t}x' \right\|_2^2.
$$

Define

$$
V(t) := \mathbb{E}_{x\sim p_t} \left\| x - \mathbb{E}_{x'\sim p_t}x' \right\|_2^2, \quad \mu(t) := \mathbb{E}_{x\sim p_t}[x].
$$

Next we calculate the derivate of $V(t)$. First, by applying Itô's formula to $\|x(t)\|^2$, we have

$$
d\|x(t)\|_2^2 = 2\langle x(t), f(x(t),u(x(t))) \rangle \, dt + \|g(x(t),u(x(t)))\|_F^2 \, dt
$$
$$
+ 2\langle x(t), g(x(t),u(x(t))) \, dw(t) \rangle.
$$

Then taking expectation for both side and using the fact $\mathbb{E}dw(t)=0$, we have

$$
\frac{d}{dt}\mathbb{E}\|x(t)\|_2^2 = 2\,\mathbb{E}\langle x(t), f(x(t),u(x(t))) \rangle + \mathbb{E}\|g(x(t),u(x(t)))\|_F^2.
$$

Next, we have

$$
\frac{d}{dt}\|\mu(t)\|_2^2 = 2\langle \mu(t), \mathbb{E}f(x(t),u(x(t))) \rangle.
$$

Then by the fact that $V(t) = \mathbb{E}\|x(t)\|_2^2 - \|\mu(t)\|_2^2$ we obtain

$$
\frac{d}{dt}V(t) = 2\,\mathbb{E}\left[ \langle x(t)-\mu(t), f(x(t),u(x(t))) - f(\mu(t),u(\mu(t))) \rangle \right] + \mathbb{E}\left[ \|g(x(t),u(x(t)))\|_F^2 \right]
$$

$$
\leq 2\,\mathbb{E}\|x(t)-\mu(t)\|_2 \cdot \|f(x(t),u(x(t))) - f(\mu(t),u(\mu(t)))\|_2 + \mathbb{E}\|g(x(t),u(x(t)))\|_F^2
$$

$$
\leq 2L_f(1+L_u)V(t) + G^2,
$$

where the last inequality follows from the Lipschitz continuity of $f$ and $u$.

Applying Grönwall's lemma, we get

$$
V(t) \leq \frac{G^2}{2L_f(1+L_u)} \left( e^{2L_f(1+L_u)t} - 1 \right) \leq \frac{G^2}{2L_f(1+L_u)} \left( e^{2L_f(1+L_u)T} - 1 \right). \tag{A.2}
$$

Substituting equation A.2 into equation A.1 completes the proof. $\qquad\square$

## A.2 EXAMPLES OF CONTINUOUS-TIME DYNAMICS WITH LOW COMPLEXITY

In this section, we present several example continuous-time dynamic classes. We first consider the setting where $\mathcal{F}$ and $\mathcal{G}$ are finite. The following proposition shows that, under a *quadratic* density model, both the eluder dimension and the bracketing number of the induced class are small.

*Proof of Proposition 5.9.* **Eluder dimension.** For simplicity, denote $y := (u, x, t)$. By the definition of Hellinger distance, we have

$$\mathbb{H}^2(p_{f,g}(y)\|p_{f^*,g^*}(y)) = 1 - \int_x \sqrt{p_{f,g}(x|y) \cdot p_{f^*,g^*}(x|y)}\, dx$$

$$= 1 - \int_x \phi(y)^\top \mu_{f,g}(x)\mu_{f^*,g^*}(x)^\top \phi(y)\, dx$$

$$= 1 - \phi(y)^\top \left[\int_x \mu_{f,g}(x)\mu_{f^*,g^*}(x)^\top dx\right] \phi(y). \tag{A.3}$$

Therefore, the squared Hellinger distance is a linear function of the feature matrix $\phi(y)\phi(y)^\top \in \mathbb{R}^{d \times d}$. Since $\|\phi(y)\|_2 \leq 1$, it follows that

$$\|\phi(y)\phi(y)^\top\|_F \leq 1. \tag{A.4}$$

Now we bound the Frobenius norm of the matrix inside the integral:

$$\left\|\int_x \mu_{f,g}(x)\mu_{f^*,g^*}(x)^\top dx\right\|_F \leq \int_x \|\mu_{f,g}(x)\|_2 \cdot \|\mu_{f^*,g^*}(x)\|_2\, dx$$

$$\leq \left(\int_x \|\mu_{f,g}(x)\|_2^2 dx\right)^{1/2} \left(\int_x \|\mu_{f^*,g^*}(x)\|_2^2 dx\right)^{1/2}$$

$$\leq B, \tag{A.5}$$

where the last inequality uses the Cauchy–Schwarz inequality and the assumed boundedness of the $\mu$ functions.

Putting together the bounds in equation A.3, equation A.4, and equation A.5, and invoking Proposition 19 in Liu et al. (2022) and Proposition 6 in Russo & Van Roy (2013), we obtain

$$\mathrm{DE}_1(\Psi, \mathcal{Y}, \epsilon) \leq \mathrm{DE}_2(\Psi, \mathcal{Y}, \epsilon) \lesssim d^2 \log\left(1 + \frac{B^2}{\epsilon^2}\right).$$

**Bracketing number.** We take the brackets to be $[l_1, l_2] = [p_{f,g}(x \mid y), p_{f,g}(x \mid y)]$ for each pair $(f, g)$. This collection is trivially a valid bracketing family, and therefore the $\epsilon$-bracketing number is bounded by the cardinality of the model class, i.e.,

$$N_{[]}(\epsilon, \Psi, \|\cdot\|) \leq |\mathcal{F}| \times |\mathcal{G}|.$$

$\square$

Next, we consider the setting where the classes $\mathcal{F}$ and $\mathcal{G}$ may have infinite cardinality. We show that, even in this case, the induced model class still admits a small eluder dimension and a controlled bracketing number.

*Proof of Proposition 5.10.* **Eluder dimension.** Define $\mu_{f,g} := M_{f,g}\mu$. Following the proof of Proposition 5.9, we can verify that

$$\left\|\int_x \mu_{f,g}(x)\mu_{f^*,g^*}(x)^\top dx\right\|_F \leq \int_x \|\mu_{f,g}(x)\|_2 \cdot \|\mu_{f^*,g^*}(x)\|_2\, dx$$

$$\leq \|M_{f,g}\|_F \|M_{f^*,g^*}\|_F \int_x \|\mu(x)\|_2 \cdot \|\mu(x)\|_2\, dx$$

$$\leq B^{3/2} \int_x \|\mu(x)\|_1 dx$$

$$\leq dB^{3/2},$$

where we use the fact that each $[\mu(x)]_i$ is a density function. Thus we can conclude that

$$d_{1/\epsilon} \lesssim d^2 \log\left(1 + \frac{d^2 B^3}{\epsilon^2}\right).$$

**Bracketing number.** We now construct an $\epsilon$-bracketing set. For each pair $(l_1, l_2)$, we consider brackets of the form

$$l = \left(\phi(u,x,t)^\top M_l\, \mu(x')\right)^2, \qquad M_l = ([M_l]_{i,j}) = \begin{pmatrix} k_{1,1}\,\zeta & k_{1,2}\,\zeta & \cdots & k_{1,d}\,\zeta \\ k_{2,1}\,\zeta & k_{2,2}\,\zeta & \cdots & k_{2,d}\,\zeta \\ \vdots & \vdots & \ddots & \vdots \\ k_{d,1}\,\zeta & k_{d,2}\,\zeta & \cdots & k_{d,d}\,\zeta \end{pmatrix},$$

$$\zeta := \frac{\epsilon}{3d^2 B}, \qquad k_{i,j} \in \left\{-\lceil B/\zeta\rceil,\ -\lceil B/\zeta\rceil + 1,\ \ldots,\ \lceil B/\zeta\rceil\right\} \subset \mathbb{Z}.$$

For any matrix $M$, define its upper bracket matrix $\widetilde{M}$ by $[\widetilde{M}]_{i,j} := \lceil [M]_{i,j}/\zeta\rceil \cdot \zeta$. By construction, $[\widetilde{M}]_{i,j} \geq [M]_{i,j}$, and therefore

$$(\phi(u,x,t)^\top \widetilde{M}\, \mu(x'))^2 = \left(\sum_{i,j} [\phi(u,x,t)]_{i,j}\, [\widetilde{M}]_{i,j}\, [\mu(x')]_{i,j}\right)^2$$

$$\geq \left(\sum_{i,j} [\phi(u,x,t)]_{i,j}\, [M]_{i,j}\, [\mu(x')]_{i,j}\right)^2 \qquad (A.6)$$

$$= (\phi(u,x,t)^\top M\, \mu(x'))^2.$$

We now bound the bracket width. For any $u, x, t$,

$$\int_{x'} \left|(\phi(u,x,t)^\top \widetilde{M}\, \mu(x'))^2 - (\phi(u,x,t)^\top M\, \mu(x'))^2\right| dx'$$

$$= \int_{x'} \left(\phi(u,x,t)^\top \widetilde{M}\, \mu(x') + \phi(u,x,t)^\top M\, \mu(x')\right) \left|\sum_{i,j} [\phi(u,x,t)]_{i,j}\left([\widetilde{M}]_{i,j} - [M]_{i,j}\right)[\mu(x')]_{i,j}\right| dx'$$

$$\leq \sqrt{B}\left(2\sqrt{B} + d^2\zeta^2\right)\zeta \cdot \int_{x'} \sum_{i,j} [\mu(x')]_{i,j}\, dx'$$

$$\leq d^2\sqrt{B}\left(2\sqrt{B} + d^2\zeta^2\right)\zeta$$

$$\leq \epsilon. \qquad (A.7)$$

Thus, the constructed family forms an $\epsilon$-bracketing set. Its cardinality is bounded by

$$\left(\frac{2B}{\zeta}\right)^{d\times d} = O\left(\frac{3d^2 B^3}{\epsilon}\right)^{d\times d}. \qquad (A.8)$$

$\square$

### A.3 CONSTRUCTION EXAMPLE FOR PROPOSITION 5.9

The quadratic form presented in Proposition 5.9 is well-motivated and can be constructed explicitly. For simplicity, let us consider a quadratic density function $p(y \mid t)$ that is independent of policy $u$ and state $x$. Let us assume $p(y \mid t) = (\phi(t)^\top \mu(y))^2$ with $\phi(t), \mu(y) \in \mathbb{R}^2$. Then we can take $\phi(t) = (\cos(t), \sin(t))^\top$ and $\mu(y) = (c_1 e^{-y^2}, c_2 y e^{-y^2})^\top$, where $c_1 = (2/\pi)^{1/4}$ and $c_2 = 2(2/\pi)^{1/4}$. The resulting density is

$$p(y \mid t) = \left[(2/\pi)^{1/4}\cos(t)e^{-y^2} + 2(2/\pi)^{1/4}\sin(t)y e^{-y^2}\right]^2.$$

This defines a valid, time-evolving probability density because the basis functions in $\mu(y)$ are orthonormal, satisfying $\int \mu_i(y)\mu_j(y)\, dy = \delta_{ij}$, and the coefficients in $\phi(t)$ satisfy $\cos^2(t) + \sin^2(t) = 1$, ensuring $\int p(y \mid t)\, dy = 1$ for all $t$.

The drift $f(y,t)$ and diffusion $g(y,t)$ of an SDE generating this density can be obtained from the Fokker–Planck equation $\partial_t p = -\partial_y(fp) + \frac{1}{2}\partial_y^2(g^2 p)$. Setting $g = 1$, we have

$$f(y,t) = \frac{1}{p(y \mid t)} \int_{-\infty}^{y} \left[ \frac{1}{2} \frac{\partial^2 p}{\partial z^2} - \frac{\partial p}{\partial t} \right] dz.$$

Although the resulting drift does not have a simple closed form, it can be computed explicitly given $p(y \mid t)$. In this sense, the SDE with $(f, 1)$ provides a valid example satisfying Proposition 5.9.

Additionally, though classical SDEs do not directly yield the quadratic form of Proposition 5.9, we can identify related structures in well-known processes. The classical Ornstein–Uhlenbeck (OU) process provides a case that satisfies a linear form $p(y \mid t) = \phi(t)^\top \mu(y)$. Its spectral representation (see Chapter 5.4 of Risken & Frank (1996)) is given by

$$p(y \mid t, y_0) = \sum_{n=0}^{\infty} e^{\lambda_n t} \psi_n(y) \psi_n(y_0),$$

where $\lambda_n = -n\gamma$ with $\gamma > 0$ denoting the mean-reversion rate, and $\{\psi_n(y)\}$ are the Hermite eigenfunctions of the corresponding OU generator. This representation constitutes a linear inner product in an infinite-dimensional space, illustrating that such structures arise naturally even when the SDE itself has simple drift and diffusion coefficients.

### A.4 Derivation for equation 4.1

Starting from the definition of the value function:

$$V_{f,g}(u, x, s) := \mathbb{E}_{x(\cdot) \sim p_{f,g}(u, x_{\text{ini}})} \left[ \int_{t=s}^{T} b(x(t), u(x(t)))\, dt \,\middle|\, x(s) = x \right]$$

We split the time integral at $s + \Delta$:

$$V_{f,g}(u, x, s) = \mathbb{E}_{x(\cdot) \sim p_{f,g}(u, x_{\text{ini}})} \left[ \int_{t=s}^{s+\Delta} b(x(t), u(x(t)))\, dt + \int_{t=s+\Delta}^{T} b(x(t), u(x(t)))\, dt \,\middle|\, x(s) = x \right]$$

For an Itô process, the future evolution $\{x(t)\}_{t \geq s+\Delta}$ depends only on $x(s + \Delta)$ and is conditionally independent of the past $\{x(t)\}_{t \leq s}$ given $x(s + \Delta)$. Therefore, we can apply the tower property of conditional expectation:

$$V_{f,g}(u, x, s) = \mathbb{E}_{x(\cdot) \sim p_{f,g}(u, x)} \left[ \int_{t=s}^{s+\Delta} b(x(t), u(x(t)))\, dt \right]$$
$$+ \mathbb{E}_{x' \sim p_{f,g}(u, x, \Delta)} \left[ \mathbb{E}_{x(\cdot) \sim p_{f,g}(u, x')} \left[ \int_{t=s+\Delta}^{T} b(x(t), u(x(t)))\, dt \,\middle|\, x(s + \Delta) = x' \right] \right]$$

By Markov property of the Itô SDE and the definition of the value function $V_{f,g}(u, x', s + \Delta)$, we obtain:

$$V_{f,g}(u, x, s) = \mathbb{E}_{x(\cdot) \sim p_{f,g}(u, x)} \left[ \int_{0}^{\Delta} b(x(t), u(x(t)))\, dt \right] + \mathbb{E}_{x' \sim p_{f,g}(u, x, \Delta)} \left[ V_{f,g}(u, x', s + \Delta) \right],$$

which is precisely equation 4.1.

# B    PROOF OF MAIN THEOREM

We first define several notations for convenience. Let

$$p_n^*(x,t) := p_{f^*,g^*}(u_n,x,t), \qquad p_n(x,t) := p_{f_n,g_n}(u_n,x,t),$$
$$V_n^*(x,t) := V_{f^*,g^*}(u_n,x,t), \qquad V_n(x,t) := V_{f_n,g_n}(u_n,x,t).$$

## B.1    AUXILIARY LEMMAS

The following lemma shows that the difference in expectations between two distributions can be bounded by the variance of one distribution and their Hellinger distance, which plays a key role in deriving our variance-dependent regret bound.

**Lemma B.1** (Wang et al. 2024b;a). *Let $p,q \in \Delta([0,1])$ be two probability distributions over $[0,1]$. Define the variance of $p$ as*

$$\mathrm{VaR}_p := \mathbb{E}_{x \sim p}\left[(x - \mathbb{E}_{x \sim p}[x])^2\right].$$

*Then the following inequality holds:*

$$|\mathbb{E}_{x \sim p}[x] - \mathbb{E}_{x \sim q}[x]| \lesssim \sqrt{\mathrm{VaR}_p \cdot \mathbb{H}^2(p \,\|\, q)} + \mathbb{H}^2(p \,\|\, q).$$

The following lemma provides a concentration inequality for martingale difference sequences without boundedness assumptions, which is essential for handling heavy-tailed or unbounded noise in our analysis.

**Lemma B.2** (Unbounded Freedman's inequality, Dzhaparidze & Van Zanten (2001); Fan et al. (2017)). *Let $\{x_i\}_{i=1}^n$ be a stochastic process adapted to a filtration $\{\mathcal{G}_i\}_{i=1}^n$, where $\mathcal{G}_i = \sigma(x_1,\ldots,x_i)$. Suppose $\mathbb{E}[x_i \mid \mathcal{G}_{i-1}] = 0$ and $\mathbb{E}[x_i^2 \mid \mathcal{G}_{i-1}] < \infty$ almost surely. Then, for any $a,v,y > 0$, we have*

$$\mathbb{P}\left(\sum_{i=1}^n x_i > a, \ \sum_{i=1}^n \left(\mathbb{E}[x_i^2 \mid \mathcal{G}_{i-1}] + x_i^2 \cdot \mathbb{1}\{|x_i| > y\}\right) < v^2\right) \leq \exp\left(\frac{-a^2}{2(v^2 + ay/3)}\right).$$

*Equivalently, with probability at least $1 - \delta$, the following high-probability bound holds:*

$$\sum_{i=1}^n x_i \leq \sqrt{2 \sum_{i=1}^n \left(\mathbb{E}[x_i^2 \mid \mathcal{G}_{i-1}] + x_i^2 \cdot \mathbb{1}\{|x_i| > y\}\right) \log(1/\delta)} + \frac{y}{3}\log(1/\delta).$$

The next lemma bounds the sum of truncated random variables in terms of their conditional expectations, which is useful for controlling tail contributions in martingale-adapted processes.

**Lemma B.3** (Lemma 8, Zhang et al. 2022). *Let $\{x_i\}_{i=1}^n$ be a nonnegative stochastic process adapted to a filtration $\{\mathcal{G}_i\}_{i \geq 1}$, i.e., $x_i \geq 0$ almost surely. Then, for any $\delta \in (0,1)$, with probability at least $1 - \delta$, we have*

$$\sum_{i=1}^n \min\{x_i, y\} \leq 4y \log(4/\delta) + 4\log(4/\delta)\sum_{i=1}^n \mathbb{E}[x_i \mid \mathcal{G}_{i-1}].$$

We also include the following two auxiliary lemmas that will be used in our analysis.

**Lemma B.4** (Lemma 11, Wang et al. 2024b). *Let $G > 0$ and $a < G/2$ be positive constants. Let $\{C_i\}_{i=0}^M$ be a sequence of positive real numbers, where $M = \lceil \log_2(H/G) \rceil$, satisfying:*

- $C_i \leq 2^i G + \sqrt{aC_{i+1}} + a$ for all $i \geq 0$;
- $C_i \leq H$ for all $i \geq 0$, where $H > 0$ is a positive constant.

*Then it holds that $C_0 \leq 4G$.*

**Lemma B.5.** *For any random variable $X \in [0,1]$, we have $\mathrm{Var}(X^2) \leq 4\mathrm{Var}(X)$.*

*Proof.* Let $Y$ be an independent copy of $X$. Then,

$$\mathrm{Var}(X^2) = \tfrac{1}{2}\mathbb{E}[(X^2 - Y^2)^2] = \tfrac{1}{2}\mathbb{E}[(X-Y)^2(X+Y)^2] \leq \tfrac{1}{2} \cdot 4\,\mathbb{E}[(X-Y)^2] = 2\,\mathbb{E}[(X-Y)^2],$$

where we used $(X+Y)^2 \leq 4$ since $X,Y \in [0,1]$.

Next, we observe:

$$\mathbb{E}[(X - Y)^2] = \mathbb{E}[X^2] + \mathbb{E}[Y^2] - 2\,\mathbb{E}[X]\,\mathbb{E}[Y] = 2\,\mathbb{E}[X^2] - 2\,\mathbb{E}[X]^2 = 2\,\mathrm{Var}(X),$$

where the last equality follows from $\mathbb{E}[X] = \mathbb{E}[Y]$ and $\mathbb{E}[X^2] = \mathbb{E}[Y^2]$. Combining both steps, we get

$$\mathrm{Var}(X^2) \le 2 \cdot 2\,\mathrm{Var}(X) = 4\,\mathrm{Var}(X),$$

which concludes the proof. □

## B.2 Lemmas on confidence sets

In this section we prove several lemmas about confidence sets established in Algorithm 1. We first introduce several technical lemmas.

**Lemma B.6** (Lemma E.2, Wang et al. 2023). *Let $p_1 : \mathcal{Y} \to \Delta(\mathcal{X})$ and $p_2 : \mathcal{Y} \times \mathcal{X} \to \mathbb{R}_+$ satisfying $\sup_{y \in \mathcal{Y}} \int_x p_2(y, x)\, dx \le s$, then for any distribution $\mathcal{D} \in \Delta(\mathcal{Y})$, we have*
$$\mathbb{E}_{y \sim \mathcal{D}}\left[H^2(p_1(y) \,\|\, p_2(y, \cdot))\right] \le (s - 1) - 2\log \mathbb{E}_{y \sim \mathcal{D},\, x \sim p_1(y)} \exp\left(-\tfrac{1}{2} \log\left(p_1(y, x)/p_2(y, x)\right)\right).$$

**Lemma B.7** (Lemma E.3, Wang et al. 2023). *Let $\Upsilon$ be a class of conditional distributions. Consider a dataset $D = \{y_i, x_i\}_{i=1}^n$ generated as follows: each $y_i \sim \mathcal{D}_i$, where $\mathcal{D}_i$ may depend on the past history $(y_{1:i-1}, x_{1:i-1})$, and each $x_i$ is drawn according to the ground-truth conditional distribution $p^\star(y_i, \cdot)$. Fix $\delta \in (0, 1)$. Then, with probability at least $1 - \delta$, for every $p \in \Upsilon$ we have*

$$\sum_{i=1}^n \mathbb{E}_{y \sim \mathcal{D}_i}\left[\mathbb{H}^2(p(y, \cdot) \,\|\, p^\star(y, \cdot))\right] \le 6n\epsilon \;+\; 2\sum_{i=1}^n \log\left(\frac{p^\star(y_i, x_i)}{p(y_i, x_i)}\right) \;+\; 8\log\left(\frac{\mathcal{N}_{[]}(\epsilon, \Upsilon, \|\cdot\|)}{\delta}\right).$$

*Here, $\mathcal{N}_{[]}(\epsilon, \Upsilon, \|\cdot\|)$ denotes the $\epsilon$-bracketing number defined in Definition 5.6. Moreover, rearranging the above inequality yields*

$$\sum_{i=1}^n \log\left(\frac{p(y_i, x_i)}{p^\star(y_i, x_i)}\right) \;\le\; 3n\epsilon \;+\; 4\log\left(\frac{N_{[]}(\epsilon, \Upsilon, \|\cdot\|)}{\delta}\right).$$

*Proof.* First, let $\widetilde{\Upsilon}$ denote an $\epsilon$-bracketing of $\Upsilon$. Applying Lemma 24 of Agarwal et al. (2020) to the function class $\widetilde{\Upsilon}$ and using the Chernoff method, we obtain that, with probability at least $1 - \delta$, for all $\widetilde{p} \in \widetilde{\Upsilon}$,

$$\underbrace{-\log \mathbb{E}_{D'} \exp(L(\widetilde{p}(D), D'))}_{(i)} \;\le\; \underbrace{-L(\widetilde{p}(D), D) + 2\log\left(\mathcal{N}_{[]}(\epsilon, \Upsilon, \|\cdot\|)/\delta\right)}_{(ii)}.$$

Next, fix any $p \in \Upsilon$ and choose $\widetilde{p} \in \widetilde{\Upsilon}$ to be its upper bracket (i.e., $p \le \widetilde{p}$). Set

$$L(p, D) = \sum_{i=1}^n -\tfrac{1}{2}\log(p^\star(y_i, x_i)/p(y_i, x_i)).$$

Then the right-hand side of (ii) becomes

$$(ii) = \tfrac{1}{2}\sum_{i=1}^n \log(p^\star(y_i, x_i)/\widetilde{p}(y_i, x_i)) + 2\log\left(\mathcal{N}_{[]}(\epsilon, \Upsilon, \|\cdot\|)/\delta\right)$$

$$\le \tfrac{1}{2}\sum_{i=1}^n \log(p^\star(y_i, x_i)/p(y_i, x_i)) + 2\log\left(\mathcal{N}_{[]}(\epsilon, \Upsilon, \|\cdot\|)/\delta\right).$$

Since $\mathbb{H}$ is a metric,

$$\sum_{i=1}^n \mathbb{E}_{y \sim \mathcal{D}_i} \mathbb{H}^2(p(y, \cdot),\, p^\star(y, \cdot)) \le \sum_{i=1}^n \mathbb{E}_{y \sim \mathcal{D}_i}\left(\mathbb{H}(p(y, \cdot), \widetilde{p}(y, \cdot)) + \mathbb{H}(\widetilde{p}(y, \cdot), p^\star(y, \cdot))\right)^2$$

$$\le 2\underbrace{\sum_{i=1}^n \mathbb{E}_{y \sim \mathcal{D}_i} \mathbb{H}^2(p(y, \cdot), \widetilde{p}(y, \cdot))}_{(iii)} + 2\underbrace{\sum_{i=1}^n \mathbb{E}_{y \sim \mathcal{D}_i} \mathbb{H}^2(\widetilde{p}(y, \cdot), p^\star(y, \cdot))}_{(iv)}.$$

By Definition 5.6, $\int_x |\widetilde{p}(y,x) - p(y,x)| \leq \epsilon$ for all $y$. Hence,

$$(iii) = \sum_{i=1}^n \mathbb{E}_{y \sim \mathcal{D}_i} \mathbb{H}^2(p(y,\cdot), \widetilde{p}(y,\cdot))$$

$$\leq \sum_{i=1}^n \mathbb{E}_{y \sim \mathcal{D}_i} 2 \int_x |p(y,x) - \widetilde{p}(y,x)| \, dx$$

$$\leq 2n\epsilon.$$

Apply Lemma B.6 with $p_1 = p^\star$ and $p_2 = \widetilde{p}$. Note that

$$\sup_{y \in \mathcal{Y}} \int_x \widetilde{p}(y,x) \leq \sup_{y \in \mathcal{Y}} \int_x p(y,x) + \sup_{y \in \mathcal{Y}} \int_x |p(y,x) - \widetilde{p}(y,x)| \leq 1 + \epsilon.$$

Setting $s = 1 + \epsilon$, we obtain

$$(iv) = n\epsilon - 2\sum_{i=1}^n \log \mathbb{E}_{y,x \sim p^\star(y,\cdot)} \exp\left(-\tfrac{1}{2}\log(p^\star(y,x)/\widetilde{p}(y,x))\right)$$

$$= n\epsilon - 2\sum_{i=1}^n \log \mathbb{E}_{y \sim \mathcal{D}_i} \exp\left(-\tfrac{1}{2}\log(p^\star(y,x_i)/\widetilde{p}(y,x_i))\right)$$

$$= n\epsilon - 2\log \mathbb{E}_{y,x \sim D'}\left[\exp\left(\sum_{i=1}^n -\tfrac{1}{2}\log(p^\star(y_i,x_i)/\widetilde{p}(y_i,x_i))\right) \Big| D\right]$$

$$= n\epsilon + 2(i).$$

Combining (iii) and (iv),

$$\sum_{i=1}^n \mathbb{E}_{y \sim \mathcal{D}_i} \mathbb{H}^2(p(y,\cdot), p^\star(y,\cdot)) \leq 6n\epsilon + 4(i).$$

Since $(i) \leq (ii)$, substituting (ii) gives

$$\sum_{i=1}^n \mathbb{E}_{y \sim \mathcal{D}_i} \mathbb{H}^2(p(y,\cdot), p^\star(y,\cdot)) \;\leq\; 6n\epsilon + 4\left[-L(\widetilde{p}(D), D) + 2\log\left(\mathcal{N}_{[]}(\epsilon, \Upsilon, \|\cdot\|)/\delta\right)\right].$$

$\square$

Then based on Lemma B.7, we now introduce lemmas about confidence set $\mathcal{P}_n$ that are instrumental for proving Theorems 5.11.

**Lemma B.8.** With probability at least $1 - \delta$, the following holds for all $n \in [N]$: $(f^*, g^*) \in \mathcal{P}_n$, and

$$\sum_{i=1}^{n-1} \sum_{k=0}^{m_i-1} \mathbb{H}^2\left(p_{f_n,g_n}(u_i, x_i(t_i^k), \Delta_{i,k}) \,\|\, p_{f^*,g^*}(u_i, x_i(t_i^k), \Delta_{i,k})\right) \leq 4\beta, \qquad (\text{B.1})$$

where $\beta = 5\log(N \cdot \mathcal{C}_{3m_N}/\delta)$. Here $\mathcal{C}_{1/\epsilon}$ denotes the shorthand notation defined in Definition 5.6.

*Proof.* We apply Lemma B.7 to the function class $\mathcal{F} \times \mathcal{G}$, using delta distributions $D_{i,k}$ centered at $(u_i, x_i(t_i^k), \Delta_{i,k})$, $p = p_{f_n,g_n}$, $\epsilon = 1/(3m_N)$. This guarantees that $(f^*, g^*) \in \mathcal{P}_n$. For equation B.1, recall that $f_n, g_n \in \mathcal{P}_n$, which guarantees

$$\sup_{(f,g) \in \mathcal{F} \times \mathcal{G}} \sum_{i=1}^{n-1} \sum_{k=0}^{m_i-1} \log\left(p_{f,g}(x_i(t_i^{k+1}) \mid u_i, x_i(t_i^k), \Delta_{i,k})/p_{f_n,g_n}(x_i(t_i^{k+1}) \mid u_i, x_i(t_i^k), \Delta_{i,k})\right) \leq \beta.$$

Therefore, by Lemma B.7, we have

$$\sum_{i=1}^{n-1} \sum_{k=0}^{m_i-1} \mathbb{H}^2\left(p_{f_n,g_n}(u_i, x_i(t_i^k), \Delta_{i,k}) \,\|\, p_{f^*,g^*}(u_i, x_i(t_i^k), \Delta_{i,k})\right)$$

$$\leq 6\boldsymbol{m}_N \epsilon + 2\beta + 8 \log\left(\frac{\mathcal{N}_{[]}(\epsilon, \Upsilon, \|\cdot\|)}{\delta}\right) \leq 4\beta. \tag{B.2}$$

$\square$

Next, we present a key lemma that uses the eluder dimension to bound the accumulated Hellinger distances.

**Lemma B.9.** Let $\mathcal{E}_{B.8}$ denote the event described in Lemma B.8. Then, under event $\mathcal{E}_{B.8}$, there exists a subset $\mathcal{N} \subseteq [N]$ such that:

- $|\mathcal{N}| \leq 13 \log^2(4\beta \boldsymbol{m}_N) \cdot d_{8\beta \boldsymbol{m}_N}$;

- For each $n \in [N]$, the indicator $n \in \mathcal{N}$ corresponds to a stopping time;

- The cumulative Hellinger distance outside $\mathcal{N}$ is bounded:

$$\sum_{i \in [N] \setminus \mathcal{N}} \sum_{k=0}^{m_i - 1} \mathbb{H}^2\left(p_{f_i, g_i}(u_i, x_i(t_i^k), \Delta_{i,k}) \,\|\, p_{f^*, g^*}(u_i, x_i(t_i^k), \Delta_{i,k})\right) \leq 3 d_{\boldsymbol{m}_N} + 7 d_{\boldsymbol{m}_N} \beta \log(\boldsymbol{m}_N).$$

*Proof.* We apply Lemma 6 from Wang et al. (2024b), using the distribution class $p_{f,g}$, the input space $\Pi \times \mathcal{X} \times [T]$, and the function class $\Psi$. $\square$

### B.3 LEMMAS ABOUT REGRET DECOMPOSITION

The following lemma provides a decomposition of the regret into four interpretable components based on differences between the learned and ground-truth dynamics.

**Lemma B.10** (Simulation Lemma, Agarwal et al. 2019). At episode $n$, the following decomposition holds:

$$V_n(x_{\text{ini}}, 0) - V_n^*(x_{\text{ini}}, 0) = I_{0,n} + \sum_{k=0}^{m_n - 1} \left(I_{1,n}^k + I_{2,n}^k + I_{3,n}^k + I_{4,n}^k\right),$$

where the individual terms are defined as follows:

$$I_{0,n} := \int_0^T b(x_n(t), u_n(t))\, dt - V_n^*(x_{\text{ini}}, 0),$$

$$I_{1,n}^k := \mathbb{E}_{x \sim p_n^*(x_n(t_n^k), \Delta_{n,k})} V_n(x, t_n^{k+1}) - V_n(x_n(t_n^{k+1}), t_n^{k+1}),$$

$$I_{2,n}^k := \mathbb{E}_{x(\cdot) \sim p_n^*(x_n(t_n^k))} \left[\int_0^{\Delta_{n,k}} b(x(t), u_n(t))\, dt\right] - \int_{t_n^k}^{t_n^{k+1}} b(x_n(t), u_n(t))\, dt,$$

$$I_{3,n}^k := \mathbb{E}_{x \sim p_n(x_n(t_n^k), \Delta_{n,k})} V_n(x, t_n^{k+1}) - \mathbb{E}_{x \sim p_n^*(x_n(t_n^k), \Delta_{n,k})} V_n(x, t_n^{k+1}),$$

$$I_{4,n}^k := \mathbb{E}_{x(\cdot) \sim p_n(x_n(t_n^k))} \left[\int_0^{\Delta_{n,k}} b(x(t), u_n(t))\, dt\right] - \mathbb{E}_{x(\cdot) \sim p_n^*(x_n(t_n^k))} \left[\int_0^{\Delta_{n,k}} b(x(t), u_n(t))\, dt\right].$$

*Proof.* We apply a telescoping argument over the discretization grid $\{t_n^k\}_{k=0}^{m_n}$. From the definition of the value function, we have

$$V_n(x_{\text{ini}}, 0) = \mathbb{E}_{x(\cdot) \sim p_n(x_{\text{ini}})} \left[\int_0^T b(x(t), u_n(t))\, dt\right]$$

$$= \mathbb{E}_{x(\cdot) \sim p_n(x_{\text{ini}})} \left[\int_0^{t_n^1} b(x(t), u_n(t))\, dt\right] + \mathbb{E}_{x \sim p_n(x_{\text{ini}}, \Delta_{n,0})} V_n(x, t_n^1).$$

Subtracting the realized cumulative reward yields

$$V_n(x_{\text{ini}}, 0) - \int_0^T b(x_n(t), u_n(t))\, dt$$

$$= \mathbb{E}_{x(\cdot) \sim p_n(x_{\mathrm{ini}})} \underbrace{\left[ \int_0^{t_n^1} b(x(t), u_n(t)) \, dt \right] - \int_0^{t_n^1} b(x_n(t), u_n(t)) \, dt}_{I_{2,n}^0 + I_{4,n}^0}$$

$$+ \underbrace{\mathbb{E}_{x \sim p_n(x_{\mathrm{ini}}, \Delta_{n,0})} V_n(x, t_n^1) - \mathbb{E}_{x \sim p_n^*(x_{\mathrm{ini}}, \Delta_{n,0})} V_n(x, t_n^1)}_{I_{3,n}^0}$$

$$+ \underbrace{\mathbb{E}_{x \sim p_n^*(x_{\mathrm{ini}}, \Delta_{n,0})} V_n(x, t_n^1) - V_n(x_n(t_n^1), t_n^1)}_{I_{1,n}^0}$$

$$+ V_n(x_n(t_n^1), t_n^1) - \int_{t_n^1}^T b(x_n(t), u_n(t)) \, dt. \tag{B.3}$$

By the Markov property of the Itô SDE, we have

$$\mathbb{E}_{x(\cdot) \sim p_{f,g}(u,x)} \left[ \int_{t_n^k}^{t_n^{k+1}} b(x(t), u(t)) \, dt \right] = \mathbb{E}_{x(\cdot) \sim p_{f,g}(u,x)} \left[ \int_0^{\Delta_{n,k}} b(x(t), u(t)) \, dt \right].$$

Using this identity recursively up to some $0 \le m^\dagger \le m_n$ leads to the expression

$$V_n(x_{\mathrm{ini}}, 0) - \int_0^T b(x_n(t), u_n(t)) \, dt$$

$$= \sum_{k=0}^{m^\dagger - 1} \left( I_{1,n}^k + I_{2,n}^k + I_{3,n}^k + I_{4,n}^k \right)$$

$$+ \mathbb{E}_{x(\cdot) \sim p_n(x_n(t_n^{m^\dagger}))} \underbrace{\left[ \int_{t_n^{m^\dagger}}^{t_n^{m^\dagger + 1}} b(x(t), u(t)) \, dt \right] - \int_{t_n^{m^\dagger}}^{t_n^{m^\dagger + 1}} b(x_n(t), u_n(t)) \, dt}_{I_{2,n}^{m^\dagger} + I_{4,n}^{m^\dagger}}$$

$$+ \underbrace{\mathbb{E}_{x \sim p_n(x_n(t_n^{m^\dagger}), \Delta_{n,m^\dagger})} V_n(x, t_n^{m^\dagger + 1}) - \mathbb{E}_{x \sim p_n^*(x_n(t_n^{m^\dagger}), \Delta_{n,m^\dagger})} V_n(x, t_n^{m^\dagger + 1})}_{I_{3,n}^{m^\dagger}}$$

$$+ \underbrace{\mathbb{E}_{x \sim p_n^*(x_n(t_n^{m^\dagger}), \Delta_{n,m^\dagger})} V_n(x, t_n^{m^\dagger + 1}) - V_n(x_n(t_n^{m^\dagger + 1}), t_n^{m^\dagger + 1})}_{I_{1,n}^{m^\dagger}}$$

$$+ V_n(x_n(t_n^{m^\dagger + 1}), t_n^{m^\dagger + 1}) - \int_{t_n^{m^\dagger + 1}}^T b(x_n(t), u_n(t)) \, dt. \tag{B.4}$$

Applying equation B.4 with $m^\dagger = m_n - 1$ and noting that $t_n^{m_n} = T$ and $V_n(\cdot, T) = 0$, we obtain

$$V_n(x_{\mathrm{ini}}, 0) - \int_0^T b(x_n(t), u_n(t)) \, dt = \sum_{k=0}^{m_n - 1} \left( I_{1,n}^k + I_{2,n}^k + I_{3,n}^k + I_{4,n}^k \right),$$

which completes the proof. $\qquad \square$

**Lemma B.11.** Let $I_{0,n}, I_{1,n}^k, \dots, I_{3,n}^k$ be terms introduced in Lemma B.10. Let $\widetilde{\mathcal{N}} \subseteq [N]$ be an episode index set satisfying $\widetilde{\mathcal{N}} \subseteq [N] \setminus \mathcal{N}$ and satisfying $n \in \widetilde{\mathcal{N}}$ is a stopping time. Then under event $\mathcal{E}_{B.8}$, with probability at least $1 - 4\delta$, the following bounds hold:

$$\sum_{n \in \widetilde{\mathcal{N}}} I_{0,n} \lesssim \sqrt{\log(1/\delta) \sum_{n=1}^N \mathrm{Var}_{f^*, g^*}^{u_n}} + \log(1/\delta),$$

$$\sum_{n \in \widetilde{\mathcal{N}}} \sum_{k=0}^{m_n-1} I_{1,n}^k \lesssim \sqrt{\sum_{n \in \widetilde{\mathcal{N}}} \sum_{k=0}^{m_n-1} \mathbb{V}_{x \sim p_n^*(x_n(t_n^k), \Delta_{n,k})} \left[ V_n(x, t_n^{k+1}) \right] \cdot \log(1/\delta) + \log(1/\delta)},$$

$$\sum_{n \in \widetilde{\mathcal{N}}} \sum_{k=0}^{m_n-1} I_{2,n}^k \lesssim \sqrt{\sum_{n=1}^{N} \boldsymbol{\Delta}_n^2 \log(1/\delta)},$$

$$\sum_{n \in \widetilde{\mathcal{N}}} \sum_{k=0}^{m-1} |I_{3,n}^k| \lesssim \sqrt{d_{\boldsymbol{m}_N} \beta \log(\boldsymbol{m}_N) \sum_{n \in \widetilde{\mathcal{N}}} \sum_{k=0}^{m_n-1} \left[ \mathbb{V}_{x \sim p_n^*(x_n(t_n^k), \Delta_{n,k})} V_n(x, t_n^{k+1}) \right]} + d_{\boldsymbol{m}_N} \beta \log(\boldsymbol{m}_N).$$

*Proof.* First, by Azuma-Bernstein inequality, with probability at least $1 - \delta$, we have

$$\sum_{n \in \widetilde{\mathcal{N}}} I_{0,n} = \sum_{n=1}^{N} \mathbb{1}(n \in \widetilde{\mathcal{N}}) \left( \int_{t=0}^{T} b(x_n(t), u_n(t)) dt - \mathbb{E}_{x(\cdot) \sim p_n^*(x_{\mathrm{ini}})} \left[ \int_{t=0}^{T} b(x(t), u(t)) dt \right] \right)$$

$$\lesssim \sqrt{\log(1/\delta) \sum_{n=1}^{N} \mathrm{Var}_{f^*, g^*}^{u_n} + \log(1/\delta)}.$$

Next, by definition,

$$I_{1,n}^k := \mathbb{E}_{x \sim p_n^*(x_n(t_n^k), \Delta_{n,k})} \left[ V_n(x, t_n^{k+1}) \right] - V_n(x_n(t_n^{k+1}), t_n^{k+1}).$$

Each $I_{1,n}^k$ is a zero-mean random variable whose variance is:

$$\mathbb{V}_{x \sim p_n^*(x_n(t_n^k), \Delta_{n,k})} \left[ V_n(x, t_n^{k+1}) \right] \leq 1,$$

since $V_n \in [0, 1]$.

We apply Bernstein's inequality for zero-mean, bounded ($\leq 1$) random variables. We have with probability at least $1 - \delta$,

$$\sum_{n \in \widetilde{\mathcal{N}}} \sum_{k=0}^{m_n-1} I_{1,n}^k = \sum_{n=1}^{N} \mathbb{1}(n \in \widetilde{\mathcal{N}}) \sum_{k=0}^{m_n-1} I_{1,n}^k$$

$$\lesssim \sqrt{\sum_{n=1}^{N} \mathbb{1}(n \in \widetilde{\mathcal{N}}) \sum_{k=0}^{m_n-1} \mathbb{V}_{x \sim p_n^*(x_n(t_n^k), \Delta_{n,k})} \left[ V_n(x, t_n^{k+1}) \right] \cdot \log(1/\delta) + \log(1/\delta)}.$$

$$= \sqrt{\sum_{n \in \widetilde{\mathcal{N}}} \sum_{k=0}^{m_n-1} \mathbb{V}_{x \sim p_n^*(x_n(t_n^k), \Delta_{n,k})} \left[ V_n(x, t_n^{k+1}) \right] \cdot \log(1/\delta) + \log(1/\delta)}.$$

Next, we recall the definition:

$$I_{2,n}^k := \mathbb{E}_{x(t) \sim p_n^*(x_n(t_n^k))} \left[ \int_0^{\Delta_{n,k}} b(x(t), u_n(t)) dt \right] - \int_{t_n^k}^{t_n^{k+1}} b(x_n(t), u_n(t)) dt.$$

Each $I_{2,n}^k$ is a martingale difference and satisfies $|I_{2,n}^k| \leq 2\Delta_{n,k}$, since $b(x, u) \leq 1$ by Assumption 5.1.

We apply the Azuma-Hoeffding inequality for bounded martingale differences. With probability at least $1 - \delta$:

$$\sum_{n \in \widetilde{\mathcal{N}}} \sum_{k=0}^{m_n-1} I_{2,n}^k = \sum_{n=1}^{N} \mathbb{1}(n \in \widetilde{\mathcal{N}}) \sum_{k=0}^{m_n-1} I_{2,n}^k$$

$$\lesssim \sqrt{\sum_{n=1}^{N} \mathbb{1}(n \in \widetilde{\mathcal{N}}) \sum_{k=0}^{m_n-1} \Delta_{n,k}^2 \log(1/\delta)}$$

$$\leq \sqrt{\sum_{n=1}^{N} \boldsymbol{\Delta}_n^2 \log(1/\delta)}.$$

Finally, by Assumption 5.1, the stage-wise reward is bounded in $[0, 1]$. Leveraging Lemma B.1, we can bound

$$|\mathbb{E}_{x \sim p_n(x_n(t_n^k), \Delta_{n,k})}[V_n(x, t_n^{k+1})] - \mathbb{E}_{x \sim p_n^*(x_n(t_n^k), \Delta_{n,k})}[V_n(x, t_n^{k+1})]|$$

in terms of the corresponding variance and squared Hellinger distance:

$$\sum_{n \in \widetilde{\mathcal{N}}} \sum_{k=0}^{m_n-1} |I_{3,n}^k|$$

$$= \sum_{n=1}^{N} \mathbb{1}(n \in \widetilde{\mathcal{N}}) \sum_{k=0}^{m_n-1} |\mathbb{E}_{x \sim p_n(x_n(t_n^k), \Delta_{n,k})}[V_n(x, t_n^{k+1})] - \mathbb{E}_{x \sim p_n^*(x_n(t_n^k), \Delta_{n,k})}[V_n(x, t_n^{k+1})]|$$

$$\lesssim \sum_{n=1}^{N} \mathbb{1}(n \in \widetilde{\mathcal{N}}) \sum_{k=0}^{m_n-1} \left[ \sqrt{\mathbb{V}_{x \sim p_n^*(x_n(t_n^k), \Delta_{n,k})} V_n(x, t_n^{k+1}) \cdot \mathbb{H}^2\Big(p_n^*(x_n(t_n^k), \Delta_{n,k}) \| p_n(x_n(t_n^k), \Delta_{n,k})\Big)} \right]$$

$$+ \sum_{n=1}^{N} \mathbb{1}(n \in \widetilde{\mathcal{N}}) \sum_{k=0}^{m_n-1} \left[ \mathbb{H}^2\Big(p_n^*(x_n(t_n^k), \Delta_{n,k}) \| p_n(x_n(t_n^k), \Delta_{n,k})\Big) \right]. \tag{B.5}$$

Applying the Cauchy–Schwarz inequality to equation B.5 yields

$$\sum_{n \in \widetilde{\mathcal{N}}} \sum_{k=0}^{m_n-1} |I_{3,n}^k|$$

$$\leq \sqrt{\sum_{n=1}^{N} \mathbb{1}(n \in \widetilde{\mathcal{N}}) \sum_{k=0}^{m_n-1} \left[ \mathbb{V}_{x \sim p_n^*(x_n(t_n^k), \Delta_{n,k})} V_n(x, t_n^{k+1}) \right]}$$

$$\cdot \sqrt{\sum_{n=1}^{N} \mathbb{1}(n \in \widetilde{\mathcal{N}}) \sum_{k=0}^{m_n-1} \left[ \mathbb{H}^2\Big(p_n^*(x_n(t_n^k), \Delta_{n,k}) \| p_n(x_n(t_n^k), \Delta_{n,k})\Big) \right]}$$

$$+ \sum_{n=1}^{N} \mathbb{1}(n \in \widetilde{\mathcal{N}}) \sum_{k=0}^{m_n-1} \left[ \mathbb{H}^2\Big(p_n^*(x_n(t_n^k), \Delta_{n,k}) \| p_n(x_n(t_n^k), \Delta_{n,k})\Big) \right]$$

$$\leq \sqrt{d_{\boldsymbol{m}_N} \beta \log(\boldsymbol{m}_N) \sum_{n=1}^{N} \mathbb{1}(n \in \widetilde{\mathcal{N}}) \sum_{k=0}^{m_n-1} \left[ \mathbb{V}_{x \sim p_n^*(x_n(t_n^k), \Delta_{n,k})} V_n(x, t_n^{k+1}) \right] + d_{\boldsymbol{m}_N} \beta \log(\boldsymbol{m}_N)}$$

$$= \sqrt{d_{\boldsymbol{m}_N} \beta \log(\boldsymbol{m}_N) \sum_{n \in \widetilde{\mathcal{N}}} \sum_{k=0}^{m_n-1} \left[ \mathbb{V}_{x \sim p_n^*(x_n(t_n^k), \Delta_{n,k})} V_n(x, t_n^{k+1}) \right] + d_{\boldsymbol{m}_N} \beta \log(\boldsymbol{m}_N)},$$

where the final inequality follows directly from Lemma B.9. □

### B.4 LEMMAS TO CONTROL TOTAL VARIANCE

The following lemma provides an upper bound on the cumulative variance of the value function estimates $V_n(x, t_n^{k+1})$ in terms of the variances of their reference optimal values $V_n^*(x, t_n^{k+1})$, an eluder-dimension-dependent complexity term, and an additional error term.

**Lemma B.12.** Let $\widetilde{\mathcal{N}} \subseteq [N]$ be the episode index set defined in Lemma B.11. Under the event $\mathcal{E}_{B.11}$, with probability at least $1 - \delta$, we have

$$\sum_{n \in \widetilde{\mathcal{N}}} \sum_{k=0}^{m_n-1} \mathbb{V}_{x \sim p_n^*(x_n(t_n^k), \Delta_{n,k})} V_n(x, t_n^{k+1})$$

$$\lesssim \sum_{n \in \widetilde{\mathcal{N}}} \sum_{k=0}^{m_n - 1} \mathbb{V}_{x \sim p_n^*(x_n(t_n^k), \Delta_{n,k})} V_n^*(x, t_n^{k+1}) + d_{\boldsymbol{m}_N} \beta \log(\boldsymbol{m}_N) + \sum_{n \in \widetilde{\mathcal{N}}} \sum_{k=0}^{m_n - 1} |I_{4,n}^k|.$$

*Proof.* First we have

$$\sum_{n \in \widetilde{\mathcal{N}}} \sum_{k=0}^{m_n - 1} \mathbb{V}_{x \sim p_n^*(x_n(t_n^k), \Delta_{n,k})} V_n(x, t_n^{k+1})$$

$$\leq 2 \sum_{n \in \widetilde{\mathcal{N}}} \sum_{k=0}^{m_n - 1} \mathbb{V}_{x \sim p_n^*(x_n(t_n^k), \Delta_{n,k})} V_n^*(x, t_n^{k+1})$$

$$+ 2 \sum_{n \in \widetilde{\mathcal{N}}} \sum_{k=0}^{m_n - 1} \mathbb{V}_{x \sim p_n^*(x_n(t_n^k), \Delta_{n,k})} \widehat{V}_n(x, t_n^{k+1}), \tag{B.6}$$

where $\widehat{V}_n(x, t) := V_n(x, t) - V_n^*(x, t)$. Next we focus on bound the second term. We introduce a more general high order momentum $C_i, i = 0, \ldots, \log(\boldsymbol{m}_N)$, where

$$C_i := \sum_{n \in \widetilde{\mathcal{N}}} \sum_{k=0}^{m_n - 1} \mathbb{V}_{x \sim p_n^*(x_n(t_n^k), \Delta_{n,k})} \widehat{V}_n^{2^i}(x, t_n^{k+1}).$$

Then we have

$$C_i = \sum_{n \in \widetilde{\mathcal{N}}} \sum_{k=0}^{m_n - 1} \mathbb{E}_{x \sim p_n^*(x_n(t_n^k), \Delta_{n,k})} \widehat{V}_n^{2^{i+1}}(x, t_n^{k+1}) - [\mathbb{E}_{x \sim p_n^*(x_n(t_n^k), \Delta_{n,k})} \widehat{V}_n^{2^i}(x, t_n^{k+1})]^2$$

$$= \sum_{n \in \widetilde{\mathcal{N}}} \sum_{k=0}^{m_n - 1} \mathbb{E}_{x \sim p_n^*(x_n(t_n^k), \Delta_{n,k})} \widehat{V}_n^{2^{i+1}}(x, t_n^{k+1}) - \widehat{V}_n^{2^{i+1}}(x_n(t_n^{k+1}), t_n^{k+1})$$

$$- [\mathbb{E}_{x \sim p_n^*(x_n(t_n^k), \Delta_{n,k})} \widehat{V}_n^{2^i}(x, t_n^{k+1})]^2 + \widehat{V}_n^{2^{i+1}}(x_n(t_n^{k+1}), t_n^{k+1})$$

$$\leq \sum_{n \in \widetilde{\mathcal{N}}} \sum_{k=0}^{m_n - 1} \underbrace{\mathbb{E}_{x \sim p_n^*(x_n(t_n^k), \Delta_{n,k})} \widehat{V}_n^{2^{i+1}}(x, t_n^{k+1}) - \widehat{V}_n^{2^{i+1}}(x_n(t_n^{k+1}), t_n^{k+1})}_{J_{1,n,i}^k}$$

$$\underbrace{- [\mathbb{E}_{x \sim p_n^*(x_n(t_n^k), \Delta_{n,k})} \widehat{V}_n^{2^i}(x, t_n^{k+1})]^2 + \widehat{V}_n^{2^{i+1}}(x_n(t_n^k), t_n^k)}_{J_{2,n,i}^k}, \tag{B.7}$$

where the last line holds since we move the index one step earlier and we use the fact $\widehat{V}_n(x, t_n^m) = 0$. Next, for $J_{1,n,i}^k$, by Azuma-Bernsetin inequality, we have with probability at least $1 - \delta$ for all $i = 0, \ldots, \log(\boldsymbol{m}_N)$,

$$\sum_{n \in \widetilde{\mathcal{N}}} \sum_{k=0}^{m_n - 1} J_{1,n,i}^k$$

$$= \sum_{n=1}^{N} \mathbb{1}(n \in \widetilde{\mathcal{N}}) \sum_{k=0}^{m_n - 1} J_{1,n,i}^k$$

$$\lesssim \sqrt{\sum_{n \in \widetilde{\mathcal{N}}} \sum_{k=0}^{m_n - 1} \mathbb{V}_{x \sim p_n^*(x_n(t_n^k), \Delta_{n,k})} \widehat{V}_n^{2^{i+1}}(x, t_n^{k+1}) \log(\log(\boldsymbol{m}_N)/\delta)} + \log(\log(\boldsymbol{m}_N)/\delta)$$

$$\leq \sqrt{C_{i+1} \log(\log(\boldsymbol{m}_N)/\delta)} + \log(\log(\boldsymbol{m}_N)/\delta). \tag{B.8}$$

For $J_{2,n,i}^k$, we have

$$J_{2,n,i}^k$$

$$= [\widehat{V}_n^{2^i}(x_n(t_n^k), t_n^k) - \mathbb{E}_{x \sim p_n^*(x_n(t_n^k), \Delta_{n,k})} \widehat{V}_n^{2^i}(x, t_n^{k+1})][\widehat{V}_n^{2^i}(x_n(t_n^k), t_n^k) + \mathbb{E}_{x \sim p_n^*(x_n(t_n^k), \Delta_{n,k})} \widehat{V}_n^{2^i}(x, t_n^{k+1})]$$

$$\leq [\widehat{V}_n^{2^i}(x_n(t_n^k), t_n^k) - [\mathbb{E}_{x \sim p_n^*(x_n(t_n^k), \Delta_{n,k})} \widehat{V}_n^{2^{i-1}}(x, t_n^{k+1})]^2][\widehat{V}_n^{2^i}(x_n(t_n^k), t_n^k) + \mathbb{E}_{x \sim p_n^*(x_n(t_n^k), \Delta_{n,k})} \widehat{V}_n^{2^i}(x, t_n^{k+1})]$$

$$\leq \prod_{j=0}^{i} [\widehat{V}_n^{2^i}(x_n(t_n^k), t_n^k) + \mathbb{E}_{x \sim p_n^*(x_n(t_n^k), \Delta_{n,k})} \widehat{V}_n(x, t_n^{k+1})] \cdot |\widehat{V}_n(x_n(t_n^k), t_n^k) - \mathbb{E}_{x \sim p_n^*(x_n(t_n^k), \Delta_{n,k})} \widehat{V}_n(x, t_n^{k+1})|$$

$$\leq 2^{i+1} |\widehat{V}_n(x_n(t_n^k), t_n^k) - \mathbb{E}_{x \sim p_n^*(x_n(t_n^k), \Delta_{n,k})} \widehat{V}_n(x, t_n^{k+1})|, \tag{B.9}$$

where we use the fact that $\mathbb{E}X^2 \geq [\mathbb{E}X]^2$. Then taking summation of equation B.9 over $n \in \widetilde{\mathcal{N}}$ and $k$, we have

$$2^{-(i+1)} \cdot \sum_{n \in \widetilde{\mathcal{N}}} \sum_{k=0}^{m_n - 1} J_{2,n,i}^k$$

$$\leq \sum_{n \in \widetilde{\mathcal{N}}} \sum_{k=0}^{m_n - 1} |\widehat{V}_n(x_n(t_n^k), t_n^k) - \mathbb{E}_{x \sim p_n^*(x_n(t_n^k), \Delta_{n,k})} \widehat{V}_n(x, t_n^{k+1})|$$

$$= \sum_{n \in \widetilde{\mathcal{N}}} \sum_{k=0}^{m_n - 1} |V_n(x_n(t_n^k), t_n^k) - V_n^*(x_n(t_n^k), t_n^k) - \mathbb{E}_{x \sim p_n^*(x_n(t_n^k), \Delta_{n,k})} V_n(x, t_n^{k+1})$$

$$+ \mathbb{E}_{x \sim p_n^*(x_n(t_n^k), \Delta_{n,k})} V_n^*(x, t_n^{k+1})|$$

$$= \sum_{n \in \widetilde{\mathcal{N}}} \sum_{k=0}^{m_n - 1} \Big| \underbrace{\mathbb{E}_{x \sim p_n(x_n(t_n^k), \Delta_{n,k})} V_n(x, t_n^{k+1}) - \mathbb{E}_{x \sim p_n^*(x_n(t_n^k), \Delta_{n,k})} V_n(x, t_n^{k+1})}_{I_{3,n}^k}$$

$$+ \underbrace{\mathbb{E}_{x(t) \sim p_n(x_n(t_n^k))} \left[ \int_{t=0}^{\Delta_{n,k}} b(x(t), u_n(t)) dt \right] - \mathbb{E}_{x(t) \sim p_n^*(x_n(t_n^k))} \left[ \int_{t=0}^{\Delta_{n,k}} b(x(t), u_n(t)) dt \right]}_{I_{4,n}^k} \Big|$$

$$\leq \sqrt{d_{\boldsymbol{m}_N} \beta \log(\boldsymbol{m}_N) \bigg( \sum_{n \in \widetilde{\mathcal{N}}} \sum_{k=0}^{m_n - 1} [\mathbb{V}_{x \sim p_n^*(x_n(t_n^k), \Delta_{n,k})} V_n(x, t_n^{k+1})] \bigg)}$$

$$+ d_{\boldsymbol{m}_N} \beta \log(\boldsymbol{m}_N) + \sum_{n \in \widetilde{\mathcal{N}}} \sum_{k=0}^{m_n - 1} |I_{4,n}^k|, \tag{B.10}$$

where the last inequality holds due to the upper bounds of $|I_{3,n}^k|$ obtained in Lemma B.11. Combining equation B.7, equation B.8 and equation B.10, we have $a < G/2$, $C_i \leq 2^i G + \sqrt{aC_{i+1}} + a$ and $C_i \leq H = \boldsymbol{m}_N$, where

$$G := \sqrt{d_{\boldsymbol{m}_N} \beta \log(\boldsymbol{m}_N) \bigg( \sum_{n \in \widetilde{\mathcal{N}}} \sum_{k=0}^{m_n - 1} [\mathbb{V}_{x \sim p_n^*(x_n(t_n^k), \Delta_{n,k})} V_n(x, t_n^{k+1})] \bigg)}$$

$$+ d_{\boldsymbol{m}_N} \beta \log(\boldsymbol{m}_N) + \sum_{n \in \widetilde{\mathcal{N}}} \sum_{k=0}^{m_n - 1} |I_{4,n}^k|,$$

$$a := \log(\log(\boldsymbol{m}_N)/\delta).$$

Therefore, by Lemma B.4, we have

$$C_0 \lesssim G. \tag{B.11}$$

Finally, substituting equation B.11 back to equation B.6, we have

$$\sum_{n \in \widetilde{\mathcal{N}}} \sum_{k=0}^{m_n - 1} \mathbb{V}_{x \sim p_n^*(x_n(t_n^k), \Delta_{n,k})} V_n(x, t_n^{k+1})$$

$$\lesssim \sum_{n \in \widetilde{\mathcal{N}}} \sum_{k=0}^{m_n-1} \mathbb{V}_{x \sim p_n^*(x_n(t_n^k), \Delta_{n,k})} V_n^*(x, t_n^{k+1}) + d_{\boldsymbol{m}_N} \beta \log(\boldsymbol{m}_N) + \sum_{n \in \widetilde{\mathcal{N}}} \sum_{k=0}^{m_n-1} |I_{4,n}^k|$$

$$+ \sqrt{d_{\boldsymbol{m}_N} \beta \log(\boldsymbol{m}_N) \left( \sum_{n \in \widetilde{\mathcal{N}}} \sum_{k=0}^{m_n-1} \left[ \mathbb{V}_{x \sim p_n^*(x_n(t_n^k), \Delta_{n,k})} V_n(x, t_n^{k+1}) \right] \right)}.$$

Using the fact that $x \lesssim \sqrt{ax} + b \Rightarrow x \lesssim a + b$, we obtain our final bound. $\qquad \square$

The following lemma bounds the cumulative variance of the optimal value function $V_n^*$ by the measurement gaps.

**Lemma B.13.** With probability at least $1 - \delta$, for all $n \in [N]$, we have

$$\sum_{k=0}^{m_n-1} \mathbb{V}_{x \sim p_n^*(x_n(t_n^k), \Delta_{n,k})} V_n^*(x, t_n^{k+1}) \lesssim \log(N/\delta) + \sqrt{\log(N/\delta) \max_{1 \leq n \leq N} \boldsymbol{\Delta}_n^2}. \tag{B.12}$$

*Proof.* Fix any $n \in [N]$. For simplicity, define $J_n := \sum_{k=0}^{m_n-1} \mathbb{V}_{x \sim p_n^*(x_n(t_n^k), \Delta_{n,k})} V_n^*(x, t_n^{k+1})$. We begin by expanding the variance:

$$J_n = \sum_{k=0}^{m_n-1} \left[ \mathbb{E}_{x \sim p_n^*(x_n(t_n^k), \Delta_{n,k})} V_n^*(x, t_n^{k+1})^2 - \left( \mathbb{E}_{x \sim p_n^*(x_n(t_n^k), \Delta_{n,k})} V_n^*(x, t_n^{k+1}) \right)^2 \right]$$

$$\leq \sum_{k=0}^{m_n-1} \Bigg\{ \underbrace{\mathbb{E}_{x \sim p_n^*(x_n(t_n^k), \Delta_{n,k})} V_n^*(x, t_n^{k+1})^2 - V_n^*(x_n(t_n^{k+1}), t_n^{k+1})^2}_{J_{1,n}^k}$$

$$+ \underbrace{V_n^*(x_n(t_n^k), t_n^k)^2 - \left( \mathbb{E}_{x \sim p_n^*(x_n(t_n^k), \Delta_{n,k})} V_n^*(x, t_n^{k+1}) \right)^2}_{J_{2,n}^k} \Bigg\}, \tag{B.13}$$

where the inequality uses the monotonicity $V_n^*(x_n(t_n^{k+1}), t_n^{k+1}) \leq V_n^*(x_n(t_n^k), t_n^k)$.

By the Azuma–Bernstein inequality, with probability at least $1 - \delta/N$,

$$\sum_{k=0}^{m_n-1} J_{1,n}^k \lesssim \sqrt{\sum_{k=0}^{m_n-1} \mathbb{V}_{x \sim p_n^*(x_n(t_n^k), \Delta_{n,k})} V_n^*(x, t_n^{k+1})^2 \cdot \log(N/\delta)} + \log(N/\delta)$$

$$\leq 2 \sqrt{\sum_{k=0}^{m_n-1} \mathbb{V}_{x \sim p_n^*(x_n(t_n^k), \Delta_{n,k})} V_n^*(x, t_n^{k+1}) \cdot \log(N/\delta)} + \log(N/\delta)$$

$$= 2\sqrt{J_n \log(N/\delta)} + \log(N/\delta), \tag{B.14}$$

where the second inequality follows from Lemma B.5.

For $J_{2,n}^k$, using equation 4.1 and the Markov property of Itô's SDE, we write

$$J_{2,n}^k = \left[ V_n^*(x_n(t_n^k), t_n^k) - \mathbb{E}_{x \sim p_n^*(x_n(t_n^k), \Delta_{n,k})} V_n^*(x, t_n^{k+1}) \right]$$

$$\cdot \left[ V_n^*(x_n(t_n^k), t_n^k) + \mathbb{E}_{x \sim p_n^*(x_n(t_n^k), \Delta_{n,k})} V_n^*(x, t_n^{k+1}) \right]$$

$$= \left[ \mathbb{E}_{x(\cdot) \sim p_n^*(x_n(t_n^k))} \int_{t=0}^{\Delta_{n,k}} b(x(t), u(t)) dt \right]$$

$$\cdot \left[ V_n^*(x_n(t_n^k), t_n^k) + \mathbb{E}_{x \sim p_n^*(x_n(t_n^k), \Delta_{n,k})} V_n^*(x, t_n^{k+1}) \right]$$

$$\lesssim \mathbb{E}_{x(\cdot) \sim p_n^*(x_n(t_n^k))} \int_{t=0}^{\Delta_{n,k}} b(x(t), u(t)) dt, \tag{B.15}$$

since $V_n^* \leq 1$. Hence, with probability at least $1 - \delta/N$, we have

$$\sum_{k=0}^{m_n-1} J_{2,n}^k \lesssim \sum_{k=0}^{m_n-1} \left\{ \int_{t_n^k}^{t_n^{k+1}} b(x_n(t), u_n(t)) dt \right.$$

$$+ \left[ \mathbb{E}_{x(\cdot) \sim p_n^*(x_n(t_n^k))} \int_0^{\Delta_{n,k}} b(x(t), u(t))dt - \int_{t_n^k}^{t_n^{k+1}} b(x_n(t), u_n(t))dt \right] \Bigg\}$$

$$\leq 1 + \sum_{k=0}^{m_n-1} \left[ \mathbb{E}_{x(\cdot) \sim p_n^*(x_n(t_n^k))} \int_0^{\Delta_{n,k}} b(x(t), u(t))dt - \int_{t_n^k}^{t_n^{k+1}} b(x_n(t), u_n(t))dt \right]$$

$$\lesssim 1 + \sqrt{\mathbf{\Delta}_n^2 \log(N/\delta)}, \tag{B.16}$$

where the second inequality follows from Assumption 5.1, and the third comes from Azuma-Hoeffding inequality using the bound $\int_{t_n^k}^{t_n^{k+1}} b(x(t), u(t))dt \leq \Delta_{n,k}$.

Substituting equation B.14 and equation B.16 into equation B.13, and replacing each individual confidence level $1 - \delta/N$ in equation B.14 and equation B.16 with $1 - \delta/(2N)$ (which does not affect the order of the bounds), we can apply a union bound to obtain an overall high-probability guarantee of $1 - \delta$. Consequently, with probability at least $1 - \delta$, for all $n \in [N]$,

$$J_n \lesssim \sqrt{J_n \log(N/\delta)} + 1 + \sqrt{\mathbf{\Delta}_n^2 \log(N/\delta)}$$

$$\Rightarrow J_n \lesssim \log(N/\delta) + \sqrt{\mathbf{\Delta}_n^2 \log(N/\delta)} \leq \log(N/\delta) + \sqrt{\log(N/\delta) \max_{1 \leq n \leq N} \mathbf{\Delta}_n^2}.$$

$$\square$$

The following lemma provides a global bound on the cumulative variance of the optimal value functions $V_n^*$ over all episodes. It shows that this quantity is controlled by the total variance and measurement gaps.

**Lemma B.14.** Under event $\mathcal{E}_{B.13}$, with probability at least $1 - \delta$, we have

$$\sum_{n=1}^{N} \sum_{k=0}^{m_n-1} \mathbb{V}_{x \sim p_n^*(x_n(t_n^k), \Delta_{n,k})} V_n^*(x, t_n^{k+1}) \lesssim \log^2(N/\delta) \left( 1 + \sum_{n=1}^{N} \text{Var}^{u_n} + \sum_{n=1}^{N} \mathbf{\Delta}_n^2 \right).$$

*Proof.* We define $J_n := \sum_{k=0}^{m_n-1} \mathbb{V}_{x \sim p_n^*(x_n(t_n^k), \Delta_{n,k})} V_n^*(x, t_n^{k+1})$ following Lemma B.13. Then by equation B.12 we have

$$J_n \lesssim \log(N/\delta) + \sqrt{\log(N/\delta) \max_{1 \leq n \leq N} \mathbf{\Delta}_n^2}.$$

Next we prove that the conditional expectation of $J_n$ can be bounded. First, following equation 4.1,

$$V_n^*(x, t) = \mathbb{E}_{x(\cdot) \sim p_n^*(x)} \left[ \int_t^T b(x(t), u(t))dt \right]$$

$$= \mathbb{E}_{x(\cdot) \sim p_n^*(x)} \left[ \int_t^{t+\Delta} b(x(t), u(t))dt \right] + \mathbb{E}_{x' \sim p_n^*(x, \Delta)} V_n^*(x', t + \Delta)$$

$$= \mathbb{E}_{x(\cdot) \sim p_n^*(x)} \left[ \int_0^{\Delta} b(x(t), u(t))dt \right] + \mathbb{E}_{x' \sim p_n^*(x, \Delta)} V_n^*(x', t + \Delta). \tag{B.17}$$

Then, we have

$$\text{Var}^{u_n} = \mathbb{E}_{x(\cdot) \sim p_n^*(x_{\text{ini}})} \left[ \sum_{k=0}^{m_n-1} \int_{t_n^k}^{t_n^{k+1}} b(x(t), u(t))dt - V_n^*(x_{\text{ini}}, 0) \right]^2$$

$$= \mathbb{E}_{x(\cdot) \sim p_n^*(x_{\text{ini}})} \left[ \sum_{k=0}^{m_n-1} \underbrace{\int_{t_n^k}^{t_n^{k+1}} b(x(t), u(t))dt + V_n^*(x_n(t_n^{k+1}), t_n^{k+1}) - V_n^*(x_n(t_n^k), t_n^k)}_{J_{n,k}} \right]^2$$

$$= \mathbb{V}_{x(\cdot) \sim p_n^*(x_{\text{ini}})} \left[ \sum_{k=0}^{m_n-1} J_{n,k} \right]$$

$$= \sum_{k=0}^{m_n-1} \mathbb{V}_{x(\cdot)\sim p_n^*(x_{\text{ini}})} \left[ J_{n,k} \right]. \tag{B.18}$$

The first equality follows immediately from the definition of variance, and the second comes from equation B.17. Next, on each subinterval $[t_n^k, t_n^{k+1}]$ we introduce the *temporal increment* $J_{n,k}$, for which, by construction, $\mathbb{E}[J_{n,k}] = \mathbb{E}[J_{n,k}|x_n(t_n^k)] = 0$, yielding the third equality. Then, $\{J_{n,k}\}_{k=0}^{m_n-1}$ is a martingale-difference sequence with respect to the natural filtration $\mathcal{F}_k = \sigma\big(x_n(t_n^0), \ldots, x_n(t_n^k)\big)$, so orthogonality implies

$$\mathbb{V}_{x(\cdot)\sim p_n^*(x_{\text{ini}})} \left[ \sum_{k=0}^{m_n-1} J_{n,k} \right] = \sum_{k=0}^{m_n-1} \mathbb{V}_{x(\cdot)\sim p_n^*(x_{\text{ini}})}[J_{n,k}].$$

Moreover, by the law of total variance, together with $\mathbb{E}[J_{n,k}|x_n(t_n^k)] = 0$, we have

$$\mathbb{V}_{x(\cdot)\sim p_n^*(x_{\text{ini}})} \left[ J_{n,k} \right]$$
$$= \mathbb{E}_{x\sim p_n^*(x_{\text{ini}}, t_n^k)} \left[ \mathbb{V}_{x(\cdot)\sim p_n^*(x)}[J_{n,k}|x] \right] + \mathbb{V}_{x(\cdot)\sim p_n^*(x)} \left[ \mathbb{E}_{x\sim p_n^*(x_{\text{ini}}, t_n^k)}[J_{n,k}|x] \right]$$
$$= \mathbb{E}_{x\sim p_n^*(x_{\text{ini}}, t_n^k)} \left[ \mathbb{V}_{x(\cdot)\sim p_n^*(x)}[J_{n,k}|x] \right]$$
$$= \mathbb{E}_{x\sim p_n^*(x_{\text{ini}}, t_n^k)} \left[ \mathbb{V}_{\substack{x(\cdot)\sim p_n^*(x), \\ x'\sim p_n^*(x,\Delta_{n,k})}} \left[ \int_0^{\Delta_{n,k}} b(x(t), u(t))dt + V_n^*(x', t_n^{k+1}) \right] \right]. \tag{B.19}$$

Furthermore, by Assumption 5.1 we have $\int_0^{\Delta_{n,k}} b(x_n(t), u_n(t))dt \leq \Delta_{n,k}$ for each $\Delta_{n,k}$. Thus,

$$\mathbb{E}_{x\sim p_n^*(x_{\text{ini}}, t_n^k)} \left[ \mathbb{V}_{\substack{x(\cdot)\sim p_n^*(x), \\ x'\sim p_n^*(x,\Delta_{n,k})}} \left[ \int_0^{\Delta_{n,k}} b(x(t), u(t))dt + V_n^*(x', t_n^{k+1}) \right] \right]$$
$$\leq 2\mathbb{E}_{x\sim p_n^*(x_{\text{ini}}, t_n^k)} \left[ \mathbb{V}_{x'\sim p_n^*(x,\Delta_{n,k})} \left[ V_n^*(x', t_n^{k+1}) \right] \right]$$
$$+ 2\mathbb{E}_{x\sim p_n^*(x_{\text{ini}}, t_n^k)} \left[ \mathbb{V}_{x(\cdot)\sim p_n^*(x)} \left[ \int_0^{\Delta_{n,k}} b(x(t), u(t))dt \right] \right] \tag{B.20}$$
$$\leq 2\mathbb{E}_{x\sim p_n^*(x_{\text{ini}}, t_n^k)} \left[ \mathbb{V}_{x'\sim p_n^*(x,\Delta_{n,k})} \left[ V_n^*(x', t_n^{k+1}) \right] \right] + 2\Delta_{n,k}^2. \tag{B.21}$$

Here, equation B.20 follows from the fact that $\text{Var}(a + b) = \text{Var}(a) + \text{Var}(b) + 2\text{Cov}(a, b) \leq \text{Var}(a) + \text{Var}(b) + 2\sqrt{\text{Var}(a) \cdot \text{Var}(b)} \leq 2\text{Var}(a) + 2\text{Var}(b)$. Summing equation B.21 over $k = 0, \ldots, m_n - 1$.
Thus we have, for each $n$,

$$\mathbb{E}[J_n|J_{n-1}, \ldots, J_1] \lesssim \text{Var}^{u_n} + \boldsymbol{\Delta}_n^2. \tag{B.22}$$

Applying Lemma B.3 to equation B.22 then yields

$$\sum_{n=1}^N \min\{J_n, y\} \lesssim y \log(1/\delta) + \log(1/\delta) \sum_{n=1}^N \mathbb{E}[J_n|J_{n-1}, \ldots, J_1]$$
$$\lesssim y \log(1/\delta) + \log(1/\delta) \sum_{n=1}^N \text{Var}^{u_n} + \log(1/\delta) \sum_{n=1}^N \boldsymbol{\Delta}_n^2. \tag{B.23}$$

Finally, we plug $y$ as the upper bound of $J_n$ in equation B.12 in equation B.23, leading to

$$\sum_{n=1}^N J_n \lesssim \log^2(N/\delta) \left( 1 + \sqrt{\max_{1\leq n\leq N} \boldsymbol{\Delta}_n^2} + \sum_{n=1}^N \text{Var}^{u_n} + \sum_{n=1}^N \boldsymbol{\Delta}_n^2 \right)$$
$$\lesssim \log^2(N/\delta) \left( 1 + \sum_{n=1}^N \text{Var}^{u_n} + \sum_{n=1}^N \boldsymbol{\Delta}_n^2 \right),$$

where for the second inequality we use the fact $\sqrt{x} \leq 1 + x$, thus completing the proof. □

The following lemma gives the final high-probability upper bound on the cumulative regret in terms of decomposition results established in previous lemmas.

**Lemma B.15.** Let $\widetilde{\mathcal{N}} \subseteq [N]$ be the episode index set defined in Lemma B.11. Under events $\mathcal{E}_{B.8}, \mathcal{E}_{B.11}, \mathcal{E}_{B.12}, \mathcal{E}_{B.13}, \mathcal{E}_{B.14}$, we have

$$\text{Regret}(N) \lesssim \log(N/\delta)\left( \sqrt{d_{\boldsymbol{m}_N} \beta \log(\boldsymbol{m}_N) \left( \sum_{n=1}^{N} \text{Var}_{f^*,g^*}^{u_n} + \sum_{n=1}^{N} \boldsymbol{\Delta}_n^2 \right)} \right.$$

$$\left. + N - |\widetilde{\mathcal{N}}| + d_{\boldsymbol{m}_N} \beta \log(\boldsymbol{m}_N) + \sum_{n \in \widetilde{\mathcal{N}}} \sum_{k=0}^{m_n-1} |I_{4,n}^k| \right).$$

*Proof.* By Lemma B.8, we have $R_{f^*,g^*}(u_n) \le R_{f_n,g_n}(u_n)$. For any $n \in \widetilde{\mathcal{N}}$, by Lemma B.10, we have

$$R_{f^*,g^*}(u_n) - R_{f^*,g^*}(u_n) \le R_{f_n,g_n}(u_n) - R_{f^*,g^*}(u_n)$$

$$\le \min\left\{ 1, \sum_{k=0}^{m-1} (I_{1,n}^k + I_{2,n}^k + I_{3,n}^k + I_{4,n}^k) + I_{0,n} \right\}.$$

Then we can bound the regret as

$$\text{Regret}(N) \lesssim N - |\widetilde{\mathcal{N}}| + \sum_{n \in \widetilde{\mathcal{N}}} \sum_{k=0}^{m-1} (I_{1,n}^k + I_{2,n}^k + I_{3,n}^k + I_{4,n}^k) + I_{0,n}. \tag{B.24}$$

From Lemma B.11, we have

$$\sum_{n \in \widetilde{\mathcal{N}}} \left( I_{0,n} + \sum_{k=0}^{m-1} I_{1,n}^k + I_{2,n}^k + I_{3,n}^k + I_{4,n}^k \right)$$

$$\lesssim d_{\boldsymbol{m}_N} \beta \log(\boldsymbol{m}_N) + \sqrt{d_{\boldsymbol{m}_N} \beta \log(\boldsymbol{m}_N) \sum_{n \in \widetilde{\mathcal{N}}} \sum_{k=0}^{m-1} \mathbb{V}_{x \sim p_n^*(x_n(t_n^k), \Delta)} \left[ V_n(x, t_n^{k+1}) \right]}$$

$$+ \log(1/\delta)\left( \sqrt{\sum_{n=1}^{N} \text{Var}_{f^*,g^*}^{u_n}} + \sqrt{\sum_{n=1}^{N} \boldsymbol{\Delta}_n^2} \right) + \sum_{n \in \widetilde{\mathcal{N}}} \sum_{k=0}^{m_n-1} |I_{4,n}^k|. \tag{B.25}$$

From Lemma B.12, we have

$$\sum_{n \in \widetilde{\mathcal{N}}} \sum_{k=0}^{m_n-1} \mathbb{V}_{x \sim p_n^*(x_n(t_n^k), \Delta_{n,k})} V_n(x, t_n^{k+1})$$

$$\lesssim \sum_{n \in \widetilde{\mathcal{N}}} \sum_{k=0}^{m_n-1} \mathbb{V}_{x \sim p_n^*(x_n(t_n^k), \Delta_{n,k})} V_n^*(x, t_n^{k+1}) + d_{\boldsymbol{m}_N} \beta \log(\boldsymbol{m}_N) + \sum_{n \in \widetilde{\mathcal{N}}} \sum_{k=0}^{m_n-1} |I_{4,n}^k|$$

$$\lesssim \log^2(N/\delta)\left( 1 + \sum_{n=1}^{N} \text{Var}_{f^*,g^*}^{u_n} + \sum_{n=1}^{N} \boldsymbol{\Delta}_n^2 \right) + d_{\boldsymbol{m}_N} \beta \log(\boldsymbol{m}_N) + \sum_{n \in \widetilde{\mathcal{N}}} \sum_{k=0}^{m_n-1} |I_{4,n}^k|, \tag{B.26}$$

where the second inequality holds due to Lemma B.14. Substituting equation B.26 into equation B.25, then substituting them into equation B.24, we have our final regret bound. $\square$

### B.5 PROOF OF THEOREM 5.11

We first have our concentration lemma.

**Lemma B.16.** With probability at least $1 - \delta$, we have for all $n \in [N]$, $(f^*, g^*) \in \mathcal{P}_n \cap \widehat{\mathcal{P}}_n$, and

$$\sum_{i=1}^{n-1} \sum_{k=0}^{m_i-1} \mathbb{H}^2(p_{f_n,g_n}(u_i, x_i(t_i^k), \widehat{\Delta}_{i,k}) \| p_{f^*,g^*}(u_i, x_i(t_i^k), \widehat{\Delta}_{i,k})) \le \beta, \tag{B.27}$$

where $\beta = 5 \log(N \cdot \mathcal{C}_{1/\epsilon}/\delta)$.

*Proof.* By Lemma B.8, we already have $(f^*, g^*) \in \mathcal{P}_n$ with probability at least $1 - \delta/2$. Then we apply Lemma B.12 again with $D_{i,k}$ being the delta distribution at $(u_i, x_i(t_i^k), \widehat{\Delta}_{i,k})$ guarantees $(f^*, g^*) \in \widehat{\mathcal{P}}_n$ and equation B.27 holds with probability at least $1 - \delta/2$. Taking a union bound over the two events, we conclude that with probability at least $1 - \delta$, both statements hold simultaneously. $\quad\square$

Next we have the following lemma.

**Lemma B.17.** Let the event $\mathcal{E}_{B.16}$ be the event of Lemma B.16. Then under event $\mathcal{E}_{B.16}$, there exists a set $\mathcal{N}_1 \subseteq [N]$ such that

- We have $|\mathcal{N}_1| \leq 13 \log^2(4\beta \boldsymbol{m}_N) \cdot d_{8\beta \boldsymbol{m}_N}$.

- For any $n \in [N]$, $n \in \mathcal{N}_1$ is a stopping time.

- We have

$$\sum_{i \in [N] \setminus \mathcal{N}_1} \sum_{k=0}^{m_i - 1} \mathbb{H}^2 \left( p_{f_i, g_i}(u_i, x_i(t_i^k), \widehat{\Delta}_{i,k}) \| p_{f^*, g^*}(u_i, x_i(t_i^k), \widehat{\Delta}_{i,k}) \right) \leq 3 d_{\boldsymbol{m}_N} + 7 d_{\boldsymbol{m}_N} \beta \log(\boldsymbol{m}_N).$$

*Proof.* We apply Lemma 6 in Wang et al. (2024b) here with the distribution class $p_{f,g}$, input space $\Pi \times \mathcal{X} \times [T]$ and function class $\Psi$. $\quad\square$

Next we bound $\sum_{n \in \widetilde{\mathcal{N}}} \sum_{k=0}^{m-1} |I_{4,n}^k|$ with the help of Lemma B.16 and Lemma B.17.

**Lemma B.18.** Let $\widetilde{\mathcal{N}} \subseteq [N]$ be an episode index set satisfying $\widetilde{\mathcal{N}} \subseteq [N] \setminus \mathcal{N}_1$. Under event $\mathcal{E}_{B.16}$, with probability at least $1 - \delta$, the quantities $I_{4,n}^k$ introduced in introduced in Lemma B.10 satisfy

$$\sum_{n \in \widetilde{\mathcal{N}}} \sum_{k=0}^{m-1} |I_{4,n}^k| \lesssim \sqrt{d_{\boldsymbol{m}_N} \beta \log(\boldsymbol{m}_N) \sum_{n=1}^{N} \boldsymbol{\Delta}_n^2}.$$

*Proof.* Fix $n \in \widetilde{\mathcal{N}}$ and $0 \leq k < m_n$. We have

$|I_{4,n}^k|$

$= \left| \mathbb{E}_{x(\cdot) \sim p_n(x_n(t_n^k))} \left[ \int_{t=0}^{\Delta_{n,k}} b(x(t), u(t)) dt \right] - \mathbb{E}_{x(\cdot) \sim p_n^*(x_n(t_n^k))} \left[ \int_{t=0}^{\Delta_{n,k}} b(x(t), u(t)) dt \right] \right|$

$\leq \int_{t=0}^{\Delta_{n,k}} \left| \mathbb{E}_{x \sim p_n^*(x_n(t_n^k), t)} b(x, u) - \mathbb{E}_{x \sim p_n(x_n(t_n^k), t)} b(x, u) \right| dt$

$\lesssim \int_{t=0}^{\Delta_{n,k}} \sqrt{\mathbb{V}_{x \sim p_n^*(x_n(t_n^k), t)} b(x, u) \mathbb{H}^2(p_n^*(x_n(t_n^k), t) \| p_n(x_n(t_n^k), t))}$

$\quad + \mathbb{H}^2(p_n^*(x_n(t_n^k), t) \| p_n(x_n(t_n^k), t)) dt$

$\lesssim \int_{t=0}^{\Delta_{n,k}} \mathbb{H}(p_n^*(x_n(t_n^k), t) \| p_n(x_n(t_n^k), t)) dt$

$= \underbrace{\Delta_{n,k} \cdot \mathbb{H}(p_n^*(x_n(t_n^k), \widehat{\Delta}_{n,k}) \| p_n(x_n(t_n^k), \widehat{\Delta}_{n,k}))}_{J_{1,n}^k}$

$\quad + \underbrace{\int_{t=0}^{\Delta_{n,k}} \mathbb{H}(p_n^*(x_n(t_n^k), t) \| p_n(x_n(t_n^k), t)) dt - \Delta_{n,k} \mathbb{H}(p_n^*(x_n(t_n^k), \widehat{\Delta}_{n,k}) \| p_n(x_n(t_n^k), \widehat{\Delta}_{n,k}))}_{J_{2,n}^k},$

where we use the fact that $b \leq 1$ and $\mathbb{H} \leq 1$. For $J_{1,n}^k$, we have:

$$\sum_{n \in \widetilde{\mathcal{N}}} \sum_{k=0}^{m_n - 1} J_{1,n}^k \leq \sqrt{\sum_{n \in \widetilde{\mathcal{N}}} \sum_{k=0}^{m_n - 1} \Delta_{n,k}^2} \cdot \sqrt{\sum_{n \in \widetilde{\mathcal{N}}} \sum_{k=0}^{m_n - 1} \mathbb{H}^2(p_n^*(x_n(t_n^k), \widehat{\Delta}_{n,k}) \| p_n(x_n(t_n^k), \widehat{\Delta}_{n,k}))}$$

$$\leq \sqrt{d_{\boldsymbol{m}_N} \beta \log(\boldsymbol{m}_N) \sum_{n=1}^{N} \boldsymbol{\Delta}_n^2}, \tag{B.28}$$

where the first inequality is by Cauchy-Schrawz inequality and the last one holds due to Lemma B.9. For $\{J_{2,n}^k\}_{n,k}$, because $\widehat{\Delta}_{n,k}$ is sampled uniformly from $[0, \Delta_{n,k}]$, the sequence $\{J_{2,n}^k\}_{n,k}$ forms a martingale difference sequence (MDS). Noting $|J_{2,n}^k| \leq 2\Delta_n^k$ we can apply Azuma-Hoeffding inequality to $J_{2,n}^k$, which infers that with probability at least $1 - \delta$,

$$\sum_{n \in \widetilde{\mathcal{N}}} \sum_{k=0}^{m_n-1} J_{2,n}^k = \sum_{n=1}^{N} \mathbb{1}(n \in \widetilde{\mathcal{N}}) \sum_{k=0}^{m_n-1} J_{2,n}^k \lesssim \sqrt{\sum_{n \in \widetilde{\mathcal{N}}} \sum_{k=0}^{m_n-1} \Delta_{n,k}^2 \log(1/\delta)} \leq \sqrt{\sum_{n=1}^{N} \boldsymbol{\Delta}_n^2 \log(1/\delta)}. \tag{B.29}$$

Therefore, from equation B.28 and equation B.29, we obtain our bound. □

Then we have our final proof of Theorem 5.11.

*Proof of Theorem 5.11.* We set $\widetilde{\mathcal{N}} = [N] \setminus (\mathcal{N} \cup \mathcal{N}_1)$. Since both $n \in \mathcal{N}, n \in \mathcal{N}_1$ are stopping time, then $\widetilde{\mathcal{N}}$ is also a stopping time. Clearly we have $\widetilde{\mathcal{N}} \subseteq [N] \setminus \mathcal{N}$ and $\widetilde{\mathcal{N}} \subseteq [N] \setminus \mathcal{N}_1$, thus we can apply both Lemma B.15 and B.18. Then substituting the bound of $\sum_{n \in \widetilde{\mathcal{N}}} \sum_{k=0}^{m_n-1} |I_{4,n}^k|$ from Lemma B.18 into Lemma B.15 and using the fact that

$$N - |\widetilde{\mathcal{N}}| \leq |\mathcal{N}| + |\mathcal{N}_1| \leq 26 \log^2(4\beta \boldsymbol{m}_N) \cdot d_{8\beta \boldsymbol{m}_N}$$

concludes our proof. Here, the second inequality holds due to the bounds of $|\mathcal{N}|$ in Lemma B.9 and $|\mathcal{N}_1|$ in Lemma B.17. □

---

**Algorithm 3** Lagrangian CT-MLE

---

**Require:** Episode number $N$, policy class $\Pi$, initial state $x_{\text{ini}}$, drift class $\mathcal{F}$, diffusion class $\mathcal{G}$, reward function $b$, planning horizon $T$, parameter $\eta$.
1: For each $n \in [N]$, determine a fixed measurement time sequence $0 = t_n^0 < \cdots < t_n^{m_n} = T$. For any $0 \le k < m_n$, denote measurement gaps $\Delta_{n,k} := t_n^{k+1} - t_n^k$, randomized measurement gap $\widehat{\Delta}_{n,k} \sim \text{Unif}(0, \Delta_{n,k})$.
2: **for** episode $n = 1, \ldots, N$ **do**
3:    Solve $(f_n, g_n)$ via

$$f_n, g_n = \arg\max_{(f,g) \in \mathcal{F} \times \mathcal{G}} \left\{ R_{f,g}(u_{n-1}) + \eta_n \cdot \left( \sum_{i=1}^{n-1} \sum_{k=0}^{m_i-1} \log p_{f,g}(x_i(t_i^{k+1})|u_i, x_i(t_i^k), \Delta_{i,k}) \right. \right.$$

$$\left. \left. + \sum_{i=1}^{n-1} \sum_{k=0}^{m_i-1} \log p_{f,g}(x_i(t_i^k + \widehat{\Delta}_{i,k})|u_i, x_i(t_i^k), \widehat{\Delta}_{i,k}) \right) \right\},$$

4:    Set policy $u_n$ as $u_n = \arg\max_{u \in \Pi} R_{f_n, g_n}(u)$.
5:    Execute the $n$-th episode and observe $x_n(t_n^0), x_n(t_n^0 + \widehat{\Delta}_{n,0}) \ldots, x_n(t_n^{m_n-1} + \widehat{\Delta}_{n,m_n-1}), x_n(t_n^{m_n})$.
6: **end for**
7: **return** Randomly pick an $n \in [N]$ uniformly and output $\widehat{u}$ as $u_n$.

---

## C  NUMERICAL EXPERIMENTS

Algorithm 1 (CT-MLE) is theoretically clean and analysis-friendly, but its direct use is computationally prohibitive. The core difficulty is that it optimizes a reward $R_{f,g}(u)$ over parameters $(f, g)$ subject to *two* confidence constraints, i.e., membership in the intersection $\mathcal{P}_n \cap \widehat{\mathcal{P}}_n$. This yields a constrained program with set intersections defined by likelihood inequalities, which is generally intractable at scale.

Let

$$\mathcal{L}_{f,g}^{(n)} := \sum_{i=1}^{n-1} \sum_{k=0}^{m_i-1} \log p_{f,g}(x_i(t_i^k + \Delta_{i,k})|u_i, x_i(t_i^k), \Delta_{i,k}) \tag{C.1}$$

$$\widehat{\mathcal{L}}_{f,g}^{(n)} := \sum_{i=1}^{n-1} \sum_{k=0}^{m_i-1} \log p_{f,g}(x_i(t_i^k + \widehat{\Delta}_{i,k})|u_i, x_i(t_i^k), \widehat{\Delta}_{i,k}). \tag{C.2}$$

The CT-MLE solves

$$\max_{(f,g) \in \mathcal{F} \times \mathcal{G}} R_{f,g}(u_{n-1}) \tag{C.3}$$

$$\text{s.t.} \quad \mathcal{L}_{f,g}^{(n)} \ge \max_{(f',g') \in \mathcal{F} \times \mathcal{G}} \mathcal{L}_{f',g'}^{(n)} - \beta, \tag{C.4}$$

$$\widehat{\mathcal{L}}_{f,g}^{(n)} \ge \max_{(f',g') \in \mathcal{F} \times \mathcal{G}} \widehat{\mathcal{L}}_{f',g'}^{(n)} - \beta, \tag{C.5}$$

i.e., $(f, g)$ must lie in the $\beta$-near-optimal regions of both likelihoods.

To make the problem implementable, we replace the hard constraints by penalties via standard Lagrangian relaxation. Introducing multipliers $\eta_n, \widehat{\eta}_n \ge 0$, we obtain the unconstrained surrogate

$$\max_{(f,g) \in \mathcal{F} \times \mathcal{G}} \left\{ R_{f,g}(u_{n-1}) + \eta_n \left( \mathcal{L}_{f,g}^{(n)} - \max \mathcal{L}^{(n)} + \beta \right) + \widehat{\eta}_n \left( \widehat{\mathcal{L}}_{f,g}^{(n)} - \max \widehat{\mathcal{L}}^{(n)} + \beta \right) \right\}. \tag{C.6}$$

Since $\max \mathcal{L}^{(n)}$, $\max \widehat{\mathcal{L}}^{(n)}$, and $\beta$ are constants with respect to $(f, g)$, they do not affect the maximizer and can be dropped. For simplicity we tie the multipliers, $\eta_n = \widehat{\eta}_n$, yielding the implementation-friendly objective used in Algorithm 3:

$$\max_{(f,g) \in \mathcal{F} \times \mathcal{G}} \left\{ R_{f,g}(u_{n-1}) + \eta_n \left( \mathcal{L}_{f,g}^{(n)} + \widehat{\mathcal{L}}_{f,g}^{(n)} \right) \right\}. \tag{C.7}$$

The coefficient $\eta_n$ governs the trade-off between the task reward and adherence to high-likelihood regions defined by both data fidelities ($\mathcal{L}^{(n)}$ and $\widehat{\mathcal{L}}^{(n)}$). In effect, the relaxation converts the intractable set intersection into a soft regularizer that is straightforward to optimize with standard gradient-based methods over parameterized $(f, g)$. This surrogate serves as the entry point to our experiments, enabling a scalable approximation to CT-MLE while preserving the original constraints.

## C.1 IMPLEMENTATION DETAILS

We address several practical implementation challenges for Algorithm 3. The primary challenge is computing the conditional probability density function $p_{f,g}(x_i(t_i^{k+1}) \mid u_i, x_i(t_i^k), \Delta_{i,k})$, where $\Delta_{i,k} = t_i^{k+1} - t_i^k$. Since direct maximization of the conditional log-likelihood is infeasible due to the unknown normalizing constant of the SDE transition density, we employ continuous-time score matching (Hyvärinen & Dayan, 2005). This approach eliminates the intractable normalization term by minimizing the Fisher divergence between the model score and the data score, providing a tractable and computationally efficient surrogate for MLE (Pabbaraju et al., 2023). Following Song et al. (2020), we adopt the sliced formulation to obtain unbiased and computationally efficient estimators for the drift and diffusion parameters $(f_\theta, g_\theta)$ used in Algorithm 3.

The second challenge involves determining the optimal policy $u_n$ given the estimated drift $f$ and diffusion $g$ terms. Using the learned SDE, we generate model rollouts and implement a continuous-time actor-critic update: the critic $V_\xi$ minimizes the mean-squared temporal difference error, while the actor $u_\phi$ maximizes discounted $n$-step returns through stochastic gradient ascent. Our implementation follows deterministic policy gradients (Silver et al., 2014; Lillicrap et al., 2015) but obtains exact gradients by backpropagating through the ODE, similar to neural ODE policy evaluation in continuous time (Chen et al., 2018; Yildiz et al., 2021).

We build upon the continuous-time model-based RL framework of Yildiz et al. (2021), augmenting it with additive Gaussian noise to formulate the environment dynamics as an SDE rather than an ODE. Crucially, we replace the original dynamics learning objective with a continuous-time sliced score matching (SSM) loss (Song et al., 2020). Over each of the $N_{\text{dyn}}$ gradient updates, we perform the following steps to minimize the model loss:

$$\mathcal{L}(\theta) = \mathcal{J}_{\text{SSM}}(\theta) - \eta' \mathbb{E}[V_{f_\theta,g_\theta}^{u_\psi}(x)],$$

where $\mathcal{J}_{\text{SSM}}$ is the sliced score-matching objective, the second term biases model learning toward higher policy value, and $\eta' = \frac{1}{\eta_n \kappa}$ with $\kappa > 0$ as scale factor aligning the numerical scales of the SSM loss and the (negative) planning objective.

1. **Data Sampling:** Draw a batch of $B_{\text{dyn}}$ subsequences of length $H_{\text{dyn}}$ from the training dataset $\mathcal{D}$:

$$\left\{ \left(x_i(t_0), u_i(t_0)\right), \ldots, \left(x_i(t_{H_{\text{dyn}}}), u_i(t_{H_{\text{dyn}}})\right) \right\}_{i=1}^{B_{\text{dyn}}} \sim \mathcal{D},$$

   where $x_i(t_k)$ denotes the state at measurement time $t_k$ and $u_i(t_k) = u\big(x_i(t_k)\big)$ is the corresponding control input under policy $u$.

2. **Score Matching Computation:** For each subsequence $i$ and time step $k \in \{0, \ldots, H_{\text{dyn}} - 1\}$:

   (a) Compute the interval length: $\Delta t_i^k = t_i^{k+1} - t_i^k$.

   (b) Compute the conditional mean via ODE integration:

$$\mu_\theta^{(i,k)} = \text{ODEInt}\big(f_\theta(\cdot, u_i(t_k)), x_i(t_k), [0, \Delta t_i^k]\big),$$

   where $\text{ODEInt}(\cdot)$ denotes a numerical ODE solver (we use the Dormand-Prince RK45 integrator), $f_\theta(\cdot, u_i(t_k))$ is the learned drift network with control input $u_i(t_k)$, and $[0, \Delta t_i^k]$ is the integration interval.

   (c) Evaluate the interval covariance:

$$\Sigma_\theta^{(i,k)} = \big(g_\theta(x_i(t_k), u_i(t_k))\big)^2 \Delta t_i^k,$$

   where we square the instantaneous noise scale element-wise and multiply by the interval length to obtain the diagonal covariance matrix.

   (d) Compute the model score at the interval endpoint $x_i(t_{k+1})$:

$$s_\theta^{(i,k)} = -\big(\Sigma_\theta^{(i,k)}\big)^{-1}\big(x_i(t_{k+1}) - \mu_\theta^{(i,k)}\big).$$

(e) Estimate the sliced score matching loss using $M_{\text{proj}}$ random projections. For each Rademacher vector $v_{i,k,m} \in \{\pm 1\}^d$, compute:

$$\ell_{i,k,m} = \frac{1}{2}\|s_\theta^{(i,k)}\|^2 + v_{i,k,m}^\top \nabla_x \left[v_{i,k,m}^\top s_\theta^{(i,k)}\right]\Bigg|_{x=x_i(t_{k+1})}, \quad m = 1, \ldots, M_{\text{proj}}.$$

This provides an unbiased Monte Carlo estimate of the sliced score matching loss, combining the score energy term with its directional derivative.

(f) Aggregate the batched sliced score matching loss:

$$\mathcal{J}_{\text{SSM}}(\theta) = \frac{1}{B_{\text{dyn}} \cdot H_{\text{dyn}} \cdot M_{\text{proj}}} \sum_{i=1}^{B_{\text{dyn}}} \sum_{k=0}^{H_{\text{dyn}}-1} \sum_{m=1}^{M_{\text{proj}}} \ell_{i,k,m}.$$

3. **Planning Loss Computation:**

(a) Estimate the advantage $A(x_i(t_k), u_i(t_k))$ for each state-action pair in the batch using the current critic networks:

$$\widehat{A}_i = r_i(t_k) + \gamma V'_\psi(x_i(t_{k+1})) - Q_\psi(x_i(t_k), u_i(t_k)),$$

where $Q_\psi$ is the critic network and $V'_\psi$ is the target value function.

(b) Compute the gradient of the log-transition probability with respect to the model parameters. For a Gaussian transition model parameterized by $(\mu_\theta, \Sigma_\theta)$:

$$\nabla_\theta \log P_\theta(x_{k+1}|x_k, u_k) = \nabla_\theta \left[-\frac{1}{2}\log|\Sigma_\theta| - \frac{1}{2}(x_{k+1} - \mu_\theta)^\top \Sigma_\theta^{-1}(x_{k+1} - \mu_\theta)\right].$$

This gradient is computed efficiently using automatic differentiation on the terms calculated in Step 2(b).

(c) Form the Monte Carlo estimate of the planning gradient:

$$\nabla_\theta \mathbb{E}[V] \approx \frac{1}{B_{\text{dyn}}} \sum_{i=1}^{B_{\text{dyn}}} \widehat{A}_i \cdot \nabla_\theta \log P_\theta(x_i(t_{k+1})|x_i(t_k), u_i(t_k)).$$

4. **Combined Model Update:** Update the model parameters via gradient descent:

$$\theta \leftarrow \theta - \alpha_{\text{model}}\left(\nabla_\theta \mathcal{J}_{\text{SSM}} - \eta' \nabla_\theta \mathbb{E}[V]\right),$$

using the AdamW optimizer (Kingma, 2014; Loshchilov & Hutter, 2017).

C.2 MAIN RESULTS.

We evaluate Algorithm 3 on three classic control tasks from the Gymnasium benchmark (Brockman et al., 2016; Towers et al., 2024), comparing against two state-of-the-art continuous-time baselines: ENODE (Yildiz et al., 2021) and SAC-TaCoS (Treven et al., 2024b).

**Tasks.** We consider three environments of increasing difficulty:

- **Pendulum (Easiest):** The inverted pendulum swing-up problem is a fundamental challenge in control theory. The system consists of a pendulum attached at one end to a fixed pivot, with the other end free to move. Starting from a hanging-down position, the goal is to apply torque to swing the pendulum into an upright position, aligning its center of gravity directly above the pivot. The control space represents the torque applied to the free end, while the state space includes the pendulum's x-y coordinates and angular velocity. This environment is considered the simplest due to its continuous control space and relatively straightforward dynamics with a single degree of freedom.

- **CartPole (Medium Difficulty):** The CartPole system comprises a pole attached via an unactuated joint to a cart that moves along a frictionless track. Initially, the pole is in an upright position, and the objective is to maintain balance by applying forces to the cart in either the left or right direction. The control space determines the direction of the fixed force applied to the cart, while the state space includes the cart's position and velocity, as well as the pole's angle and angular velocity. This environment presents moderate difficulty due to its discrete action space and the need to balance an inherently unstable system with coupled dynamics.

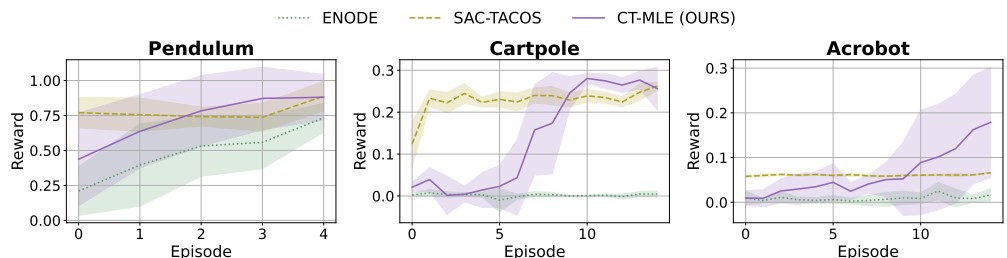

Figure 2: Performance comparison of Algorithm 3, ENODE (Yildiz et al., 2021), and SAC-TaCoS (Treven et al., 2024b) across three environments with noise $\sigma = 2.0$ ($\pm 1$ standard error).

- **Acrobot (Most Difficult):** The Acrobot system consists of two links connected in series, forming a chain with one end fixed. The joint between the two links is actuated, and the goal is to apply torques to this joint to swing the free end above a target height, starting from the initial hanging-down state. We use the fully actuated version of the Acrobot environment, as no method has successfully solved the underactuated balancing problem, consistent with Zhong & Leonard (2020). The control space is discrete and deterministic, representing the torque applied to the actuated joint, while the state space consists of the two rotational joint angles and their angular velocities. This environment is the most challenging due to its complex nonlinear dynamics involving two coupled pendulums, requiring sophisticated control strategies to coordinate the motion of both links.

For the stochastic setting, we follow Treven et al. (2024b) and inject Gaussian noise $\mathcal{N}(0, \sigma^2 I)$ at every time step, with $\sigma = 2.0$ used across all experiments. The noise is applied to all state components (such as angle $\theta$ and angular velocity $\dot{\theta}$), converting the otherwise deterministic systems into stochastic environments. Performance is evaluated after 5, 15, and 15 training episodes on Pendulum, CartPole, and Acrobot, respectively, consistent with standard evaluation protocols.

**Baselines.** **ENODE** learns dynamics using ensemble neural ODEs and optimizes a theoretically consistent continuous-time actor-critic, providing uncertainty-aware control without time discretization. However, it was not specifically designed for stochastic environments. **SAC-TaCoS** reformulates the continuous-time SDE control problem as an equivalent discrete-time extended MDP, where policies output both actions and their duration. This enables time-adaptive control using standard algorithms like SAC.

Regarding the measurement grid, ENODE adopts fixed, equidistant intervals following Yildiz et al. (2021), while SAC-TaCoS uses adaptive intervals as in Treven et al. (2024b). For simplicity, our method also uses equidistant intervals. We apply annealed Lagrange multipliers $\eta_n = \eta_{\text{base}}/n$ with $\eta_{\text{base}} = 4$, together with adaptive scaling $\kappa_n \propto$ SSM scale/planning scale to maintain a stable 10:1 SSM-to-planning ratio.

Our implementation follows the network architecture of ENODE (Yildiz et al., 2021). The dynamics model is an ensemble of 10 neural networks, each with three hidden layers of width 200 and ELU activations; the output layer uses no activation. The policy and critic networks are standard MLPs with architecture $[3, 200, 200, 1]$ (two hidden layers of width 200). The policy uses ReLU activations in the hidden layers and a Tanh output, while the critic uses Tanh activations in the hidden layers and a linear output layer.

**Results.** Figure 2 presents our main findings. Our CT-MLE algorithm achieves superior asymptotic performance across all three environments, demonstrating effective adaptation to stochastic dynamics. While SAC-TaCoS exhibits faster initial convergence and lower variance, our method ultimately achieves higher cumulative rewards after sufficient training.

ENODE shows consistently poor performance across all tasks, with minimal learning progress even after extended training. This degradation is expected given that ENODE was not designed for stochastic environments. The failure is evident in CartPole and Acrobot, where ENODE achieves no meaningful reward improvement.

The performance advantage of our method increases with task complexity. In Acrobot, the most challenging environment with complex nonlinear dynamics, the gap between our approach and the baselines is most pronounced. This suggests that our algorithm's ability to model and adapt to noisy dynamics becomes increasingly valuable as learning difficulty increases, making it particularly well-suited for complex stochastic control problems.

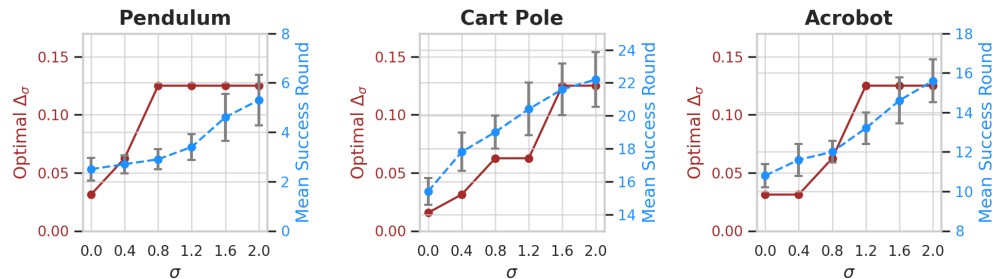

Figure 3: Optimal measurement gap $\Delta_\sigma$ and mean episodes to success ($\pm 1$ standard error) under varying environment stochasticity $\sigma$. Results averaged over 10 random seeds for Pendulum and 5 seeds for Cart Pole and Acrobot environments.

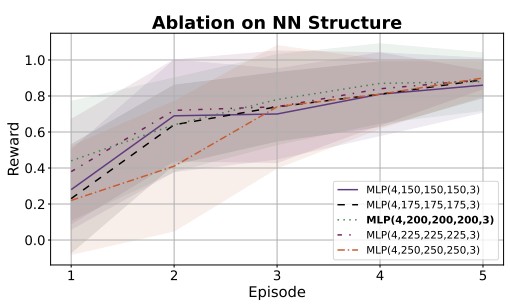

Figure 4: Ablation on Neural Network Width in the Dynamics Model

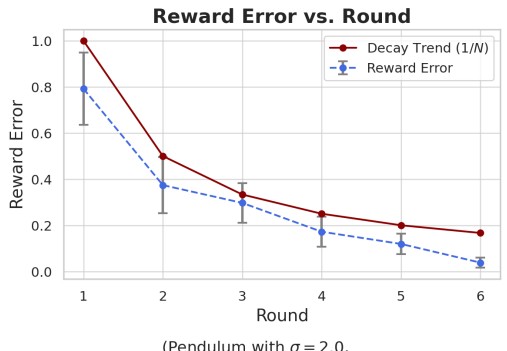

Figure 5: Reward error convergence follows the theoretical $1/\sqrt{N}$ rate (Pendulum, $\sigma = 2.0$).

## C.3 ABLATION STUDY

**Validation of Theoretical Claims.** We validate our theoretical claims through systematic numerical experiments. Following Yildiz et al. (2021), we define task success as achieving rewards of at least 0.9 after a warm-up period ($T_{\text{warm-up}} = 3$ seconds) within a planning horizon of $T = 10$ seconds. We set $\eta_n$ to a large value as it does not influence optimal gap selection. For each volatility level $\sigma \in \{0, 0.4, \ldots, 2.0\}$, we evaluate CT-MLE with equidistant measurement gaps $\Delta = 2^{-i}$ for $i \in \{0, 1, \ldots, 7\}$. The optimal gap $\Delta_\sigma$ is defined as the largest gap achieving the minimum number of episodes to success.

Figure 3 demonstrates that the optimal measurement gap $\Delta_\sigma$ increases monotonically with environment volatility $\sigma$, directly validating our theoretical analysis. This empirical observation confirms Remark 5.15, which establishes that the optimal gap for minimizing episode complexity scales proportionally with the total variance: $\Delta \propto \text{Var}^\Pi$. Higher volatility induces larger variance, necessitating wider measurement gaps for optimal performance. Notably, in low-stochasticity regimes ($\sigma \in \{0, 0.4\}$), the optimal gap is not the finest resolution tested ($2^{-7}$), confirming our theoretical prediction that excessive measurement precision yields diminishing returns.

The results further validate our algorithm's instance-dependent complexity guarantees. As shown in Figure 3, the number of episodes required for success increases with $\sigma$, confirming that our algorithm correctly identifies harder instances (higher $\sigma$) and adaptively allocates more samples. This behavior aligns with our theoretical framework, where episode complexity directly reflects the total interaction data required for convergence.

**Ablation on Neural Network Structure** To evaluate how sensitive CT-MLE is to the choice of function approximator, we conducted an ablation study on the Pendulum environment with noise level $\sigma = 2.0$, varying the network width while keeping all other components fixed. The tested architecture ranged from relatively small models with 150 hidden units per layer to wider models with up to 250 units. Across 5 random seeds, all five architectures exhibit nearly indistinguishable learning curves and achieve similar episode returns after two episodes of training (see figure 4). The smallest network performs on par with larger ones, and scaling the width beyond 200 units does

not produce meaningful improvements, suggesting that CT-MLE exhibits reasonable robustness to architectural choices.

**Numerical Convergence Rate.** We also report the reward error (mean $\pm 1$ standard error over 10 seeds) across training episodes for the Pendulum environment with $\sigma = 2.0$, using the corresponding optimal gap $\Delta_\sigma = 0.125$, as shown in Figure 5. The reward error decreases with the number of episodes $N$, and the decay trend closely follows an approximate $1/\sqrt{N}$ convergence rate, which aligns well with our theoretical predictions.

## C.4 ADDITIONAL DETAILS

All experiments were conducted on a single NVIDIA A6000 GPU. Each 15-episode Pendulum swing-up task required approximately 5 hours to complete; each 30-episode Cart Pole task required approximately 15 hours to complete; and each 25-episode Acrobot task required approximately 12 hours to complete. The peak GPU memory utilization per run ranges from 4GB to 20GB approximately. We summarize all key hyperparameters used in our experiments in Table 1, and report the neural network architecture in Table 2.

Table 1: Hyper-parameters in numerical experiment

| Hyperparameter | Default | Description |
|---|---|---|
| $\eta_{\text{base}}$ | 4 | Base value for Lagrangian Multiplier |
| $N_0$ | 3 | Number of trajectories at observation time points in initial data set |
| $H$ | 50 | Trajectory length (in seconds) in the data set |
| $N_{\text{inc}}$ | 1 | Number of trajectories at observation time points added to the data set after each episode |
| $B_{\text{dyn}}$ | 5 | Batch size of the dynamic learning |
| $N_{\text{dyn}}$ | 500 | Number of dynamic learning update iterations in each episode |
| $H_{\text{dyn}}$ | 5 | Length of each subsequence (horizon) in dynamic learning |
| $M_{\text{proj}}$ | 1 | Rademacher projections per sample in dynamic learning |
| $N_{\text{pol}}$ | 250 | Number of policy learning update iterations in each episode |
| $H_{\text{pol}}$ | 5 | Length of each subsequence (horizon) in policy learning |
| $T$ | 10 | Length of each test trajectory at the end of every episode |
| $T_{\text{warm-up}}$ | 3 | The warm-up subsequence of each test trajectory that does not collect rewards and evaluate at observation time points |
| $N_{\text{test}}$ | 10 | Number of test trajectories at the end of every episode |

Table 2: Architecture of Neural Network in numerical experiment

| Network | Architecture | Hidden Activation | Output Activation |
|---|---|---|---|
| Dynamics | [4, 200, 200, 200, 3] $\times 10$ | ELU | Linear |
| Policy | [3, 200, 200, 1] | ReLU | Tanh |
| Critic | [3, 200, 200, 1] | Tanh | Linear |

