# OpenReview forum: "Instance-Dependent Continuous-Time Reinforcement Learning via Maximum Likelihood Estimation"
_ICLR.cc/2026/Conference — Submitted to ICLR 2026_

### Official Review · Reviewer_Zkqu · 2025-10-17

**Soundness:** 3
**Presentation:** 3
**Contribution:** 3
**Rating:** 8
**Confidence:** 3

**Summary:**

This paper introduces CT-MLE, a model-based algorithm for continuous-time reinforcement learning (CTRL) that uses maximum likelihood estimation (MLE) of the state marginal density instead of directly modeling system dynamics.
The key idea is to achieve instance-dependent adaptivity, where the algorithm’s regret scales with the total reward variance rather than with fixed measurement schedules.
The authors derive theoretical guarantees, showing that regret can become independent of the measurement strategy when observation frequency adapts to problem complexity.
Additionally, they propose a randomized measurement schedule to enhance sample efficiency without additional measurement cost.

**Strengths:**

The shift from dynamic estimation to marginal density estimation via MLE is elegant and conceptually simple, offering a new angle on continuous-time RL problems.

The authors derive a variance-dependent, nearly horizon-free regret bound, marking a notable advance over existing CTRL works that rely on worst-case analysis or have exponential dependence on the time horizon.

The introduction of reward variance as a measure of instance difficulty is insightful, connecting continuous-time control with established ideas in variance-aware RL.

**Weaknesses:**

The empirical component appears secondary; the paper would benefit from more thorough experiments demonstrating practical performance, robustness, and scalability of CT-MLE.

The assumptions of finite policy, drift, and diffusion classes simplify analysis but reduce applicability to large-scale or continuous policy spaces.

**Questions:**

How sensitive is CT-MLE to the choice of function approximator in practice? Are there guidelines for selecting appropriate models?

Could the framework be extended to handle partially observable or non-stationary environments?

In what types of real-world continuous-time systems (e.g., finance, healthcare) do the authors expect the adaptive measurement mechanism to be most impactful?

---

> ### Author Response · Authors · 2025-11-17
> **Rebuttal to Reviewer Zkqu - Part 1**
>
> **Q1.** The empirical component appears secondary; the paper would benefit from more thorough experiments demonstrating practical performance, robustness, and scalability of CT-MLE.
>
>
> **A1.**  Our evaluation includes three widely used continuous-control benchmarks using the same experimental protocol as Yildiz et al. [1]. These results already indicate that CT-MLE performs reliably and robustly across standard settings. In the revised version, we have further expanded the empirical section by adding experiments that evaluate the robustness of our method with respect to different choices of function approximators, demonstrating that CT-MLE remains stable under moderate architectural variation. We would be happy to include additional experiments if there are particular aspects of performance, robustness, or scalability that the reviewer views as especially important.
>
> [1] Yildiz, C., Heinonen, M., & Lähdesmäki, H. (2021). Continuous-time model-based reinforcement learning. International Conference on Machine Learning.
>
>
>
> **Q2.** The assumptions of finite policy, drift, and diffusion classes simplify analysis but reduce applicability to large-scale or continuous policy spaces.
>
> **A2.** We appreciate the reviewer’s suggestion. In the revised manuscript we no longer assume $\Pi,\mathcal F,\mathcal G$ to be finite. All results are now stated and proved directly for general (possibly infinite) classes $\Pi,\mathcal F,\mathcal G$, with statistical complexity controlled through the bracketing number $C_{1/\epsilon}$ and the eluder dimension $d_{1/\epsilon}$.
>
> To support this change, we expanded the theoretical sections considerably:
> - We introduce the definition of bracketing number in Definition 5.6 (line 337-343) and use it to quantify the statistical complexity of the function class.
> - We modified Proposition 5.9 and added Proposition 5.10 to illustrate both finite-cardinality and genuinely infinite-cardinality families of continuous-time dynamics. Proposition 5.10 in particular provides an explicit infinite example where the dynamics class contains  infinite number of candidate transition models, and shows how $C_{1/\epsilon}$ and $d_{1/\epsilon}$ scale in this setting.
> - We added Lemmas B.6, B.7 and modified Lemma B.8 to replace the cardinality-based arguments with $\epsilon$-bracketing number. These lemmas form the backbone of the confidence-set construction used in the regret proof.
>
>
> We hope these additions make the general infinite-class setting clear and address the reviewer’s concern.

---

> > ### Author Response · Authors · 2025-11-17
> > **Rebuttal to Reviewer Zkqu - Part 2**
> >
> > **Q3.** How sensitive is CT-MLE to the choice of function approximator in practice? Are there guidelines for selecting appropriate models?
> >
> >
> > **A3.** Our implementation adopts the standard dynamics-model architecture used in prior continuous-time modeling work from [1]. Specifically, we parameterize the dynamics model $f_\theta(s,a)$ as an ensemble of 10 neural networks, each a three-layer MLP with 200 hidden units and ELU activations. To evaluate how sensitive $\text{CT-MLE}$ is to the choice of function approximator, we conducted an ablation study on the Pendulum environment with noise level $\sigma = 2.0$, varying the network width while keeping all other components fixed. The tested architectures ranged from relatively small models with 150 hidden units per layer to wider models with up to 250 units. Across 5 random seeds, all five architectures exhibit nearly indistinguishable learning curves and achieve similar episode returns after two episodes of training. The smallest network performs on par with larger ones, and scaling the width beyond 200 units does not produce meaningful improvements, suggesting that $\text{CT-MLE}$ exhibits reasonable robustness to architectural choices. All configurations attain comparable asymptotic performance (see Figure 4 in the revised manuscript and the table below).
> >
> >
> >
> > | Episode | MLP(4,150,150,150,3) | MLP(4,175,175,175,3) | MLP(4,200,200,200,3) (original) | MLP (4,225,225,225,3) | MLP (4,250,250,250,3) |
> > | ------- | -------------------- | -------------------- | ------------------------------- | --------------------- | --------------------- |
> > | Ep 1    | $0.28\pm0.22$        | $0.23\pm0.30$        | $0.44\pm 0.33$                  | $0.38\pm0.29$         | $0.22\pm0.30$         |
> > | Ep 2    | $0.69\pm0.31$        | $0.64\pm0.22$        | $0.64\pm0.26$                   | $0.72\pm0.28$         | $0.41\pm0.36$         |
> > | Ep 3    | $0.70\pm0.25$        | $0.74\pm0.19$        | $0.78\pm0.25$                   | $0.74\pm0.31$         | $0.74\pm0.34$         |
> > | Ep 4    | $0.81\pm0.23$        | $0.81\pm0.18$        | $0.87\pm0.22$                   | $0.84\pm0.20$         | $0.81\pm0.19$         |
> > | Ep 5    | $0.86\pm0.15$        | $0.89\pm0.10$        | $0.88\pm0.16$                   | $0.89\pm0.09$         | $0.90\pm0.10$         |
> >
> >
> > These findings suggest that as long as the dynamics model is sufficiently expressive to capture local system behavior, $\text{CT-MLE}$ remains robust to moderate architectural variations. In practice, we recommend starting with small ensemble networks and gradually increasing the model size to assess whether additional capacity yields further performance gains.
> >
> >
> > [1] Yildiz, C., Heinonen, M., & Lähdesmäki, H. (2021, July). Continuous-time model-based reinforcement learning. In *International Conference on Machine Learning* (pp. 12009-12018). PMLR.
> >
> >
> >
> > **Q4.** Could the framework be extended to handle partially observable or non-stationary environments?
> >
> > **A4.**  For partially observable environments, one could in principle replace state-density estimation with observation-density estimation $p(· | u, o, t)$. However, this introduces significant theoretical challenges: our regret analysis relies fundamentally on the Markov property of the state process, while observations in POMDPs are not Markovian. Extending the confidence sets and eluder-type arguments to the belief space would substantially increase both analytical and computational complexity in the continuous-time setting. Recent progress on function-approximation POMDPs [1] suggests that such extensions may be possible, but they require additional theoretical development.
> >
> > Non-stationary environments introduce an even more substantial deviation from our assumptions. Our confidence sets rely on the dynamics pair $(f^\*, g^\*)$ being fixed across episodes; this assumption is violated under drifting or time-varying dynamics. Addressing such settings would require designing time-dependent confidence sets such as sliding-window or restart-based estimators and establishing regret bounds that include variation-budget terms, following the structure used in non-stationary RL for discrete-time systems.
> >
> > In summary, both extensions are conceptually feasible but require substantial new theory tailored to the continuous-time setting. They are beyond the scope of the present work and do not detract from our current contributions; instead, they represent promising directions for future research.
> >
> >
> > [1] Liu, Q., Chung, A., Szepesvári, C., & Jin, C. (2022, June). When is partially observable reinforcement learning not scary?. In *Conference on Learning Theory* (pp. 5175-5220). PMLR.

---

> > > ### Author Response · Authors · 2025-11-17
> > > **Rebuttal to Reviewer Zkqu - Part 3**
> > >
> > > **Q5.** In what types of real-world continuous-time systems (e.g., finance, healthcare) do the authors expect the adaptive measurement mechanism to be most impactful?
> > >
> > > **A5.** Robotics is a natural setting for our adaptive measurement mechanism. Our experiments focus on three classic control environments, namely Pendulum, CartPole, and Acrobot. These systems can be regarded as simplified abstractions of real robotic tasks, so the extension is conceptually straightforward. Moreover, robotics inherently aligns with the motivation behind adaptive measurement: sensing is costly, bandwidth is limited, and the dynamics often alternate between low-variance phases (e.g., free-space motion) and high-variance phases (e.g., contact or disturbance). As discussed in Remark 5.14, adapting measurement gaps to variance provides an immediate benefit in such settings. We therefore expect robotic control to be the domain where our method has the strongest real-world impact.

---

### Official Review · Reviewer_LDX4 · 2025-10-20

**Soundness:** 3
**Presentation:** 2
**Contribution:** 2
**Rating:** 4
**Confidence:** 3

**Summary:**

This paper analyzes the continuous-time RL setting where the dynamics is modelled as an SDE with both a drift and a diffusion terms. In this setting, the authors present an algorithm for minimizing the regret during interaction with the environment. Crucially, the algorithm is based on constructing two confidence sets around the max likelihood estimate of the dynamics, and then acting optimistically w.r.t. them. The paper next provides a theoretical analysis of the algorithm.

**Strengths:**

- The continuous time RL problem is both interesting and important for the RL community as it can tackle more realistic settings than standard MDPs
- The paper provides a theoretical analysis of the proposed algorithm, proving that its regret grows only logarithmically.

**Weaknesses:**

- The algorithm cannot be implemented in practice for meaningful and general classes of dynamics F,G and policies Pi. The authors address this in Appendix C with a relaxation of the considered optimization problems. However, this makes the algorithm loose its theoretical properties.
- The analysis assumes Pi,F,G to be finite. Although the authors mention in Remark 5.2 that the analysis can be extended to more general sets using the same arguments as in a previous work (Wang et al. 2024b), they should at least add an appendix where they derive these results explicitly. Indeed, unless the theoretical analysis of this work is merely copied by (Wang et al. 2024b), it is better to add the proof for the more interesting setting explicitly.

**Questions:**

- I do not get Figure 1. I mean, in the text you criticize equidistant measurements, but here all the three options regard equidistant measurements, and what changes is just the frequency of the measurements
- where does Eq. (4.1) comes from? I would expect a formal proof for it, where you show (or at least reference and formally present) the Markov property of the Ito process, and all the passages to obtain it.
- why the regret bound does not depend on the size of the action space?

typos:
- lines 101-104: Something is missing here
- definitions/propositions/theorems should be all italics

---

> ### Author Response · Authors · 2025-11-17
> **Rebuttal to Reviewer LDX4 - Part 1**
>
> **Q1.** The algorithm cannot be implemented in practice for meaningful and general classes of dynamics $\mathcal F,\mathcal G$ and policies $\Pi$. The authors address this in Appendix C with a relaxation of the considered optimization problems. However, this makes the algorithm loose its theoretical properties.
>
> **A1.** We appreciate the reviewer’s concern. We would like to clarify that Algorithm 1 is fully implementable when $\mathcal F$, $\mathcal G$, and $\Pi$ are finite, and our guarantees directly apply to this setting. The difficulty arises only for highly expressive or infinite classes such as neural networks. In this regime, the bottleneck is the global maximization in Line 5 of Algorithm 1 (and Line 3 of Algorithm 3), which indeed becomes a hard nonconvex optimization problem. Importantly, this issue is not specific to our method. For general nonlinear function classes, including even small neural networks, finding a global maximizer is known to be NP-hard [1]. Thus, no algorithm can efficiently solve the corresponding optimization problem for all rich function classes in full generality. Our framework inherits this fundamental hardness rather than creating a new one.
>
> The primary goal of our theory is to characterize how the optimal measurement schedule should depend on the variance induced by the diffusion dynamics. This structural insight is independent of the particular solver used for the optimization subproblems and is validated empirically in Figure 3. We believe this separation between structural theory and implementable relaxations is standard in the RL theory literature, and our contribution should be interpreted in this light.
>
>
> [1] Blum, A., & Rivest, R. (1988). Training a 3-node neural network is NP-complete. _Advances in neural information processing systems_, _1_.
>
>
>
> **Q2.** The analysis assumes $\Pi,\mathcal F,\mathcal G$ to be finite. Although the authors mention in Remark 5.2 that the analysis can be extended to more general sets using the same arguments as in a previous work (Wang et al. 2024b), they should at least add an appendix where they derive these results explicitly. Indeed, unless the theoretical analysis of this work is merely copied by (Wang et al. 2024b), it is better to add the proof for the more interesting setting explicitly.
>
> **A2.** In the revised manuscript we no longer assume $\Pi,\mathcal F,\mathcal G$ to be finite. All results are now stated and proved directly for general (possibly infinite) classes $\Pi,\mathcal F,\mathcal G$, with statistical complexity controlled through the bracketing number $C_{1/\epsilon}$ and the eluder dimension $d_{1/\epsilon}$.
>
> To support this change, we expanded the theoretical sections considerably:
> - We introduce the definition of bracketing number in Definition 5.6 (line 337-343) and use it to quantify the statistical complexity of the function class.
> - We modified Proposition 5.9 and added Proposition 5.10 to illustrate both finite-cardinality and genuinely infinite-cardinality families of continuous-time dynamics. Proposition 5.10 in particular provides an explicit infinite example where the dynamics class contains infinite number of candidate transition models, and shows how $C_{1/\epsilon}$ and $d_{1/\epsilon}$ scale in this setting.
> - We added Lemmas B.6, B.7 and modified Lemma B.8 to replace the cardinality-based arguments with $\epsilon$-bracketing number. These lemmas form the backbone of the confidence-set construction used in the regret proof.
>
> We hope these additions make the general infinite-class setting clear and address the reviewer’s concern.
>
>
>
> **Q3.** I do not get Figure 1. I mean, in the text you criticize equidistant measurements, but here all the three options regard equidistant measurements, and what changes is just the frequency of the measurements
>
> **A3.** We thank the reviewer for pointing out the source of confusion. Our intention is not to criticize equidistant measurements per se, but rather to highlight that using a *fixed* measurement gap $\Delta_{n,k}$ across time as commonly done in discrete-time settings, is generally suboptimal in the continuous-time regime. In continuous dynamics, the optimal choice of $\Delta_{n,k}$ must adapt to the variance level $\mathrm{Var}^{u_n}$ induced by the diffusion.
>
> Figure 1 aims to illustrate exactly this phenomenon. Although the three measurement schedules shown (optimal / excessive / inadequate) are all equidistant for visualization purposes, they differ in their frequency, which directly determines the scale of $\sum_k \Delta_{n,k}^2$. The figure is meant to emphasize that achieving the correct alignment between $\sum_k \Delta_{n,k}^2$ and $\mathrm{Var}^{u_n}$ is essential for controlling the regret, as formalized in Theorem 5.10 and Remark 5.14. We have revised our description to reflect ths intuition.

---

> ### Author Response · Authors · 2025-11-17
> **Rebuttal to Reviewer LDX4 - Part 2**
>
> **Q4.** Where does Eq. (4.1) comes from? I would expect a formal proof for it, where you show (or at least reference and formally present) the Markov property of the Ito process, and all the passages to obtain it.
>
> **A4.** We derive Eq. (4.1) rigorously using the Markov property of Itô processes. Taking expectation over $x(\cdot) \sim p_{f,g}(u, x_{\mathrm{ini}})$, we start from the definition of the value function:
>
> $$V_{f,g}(u, x, s) := \mathbb{E} \left[ \int_{t=s}^T b(x(t), u(x(t)))  dt \bigg| x(s) = x \right]$$
>
> Splitting the time integral at $s + \Delta$:
>
> $$V_{f,g}(u, x, s) = \mathbb{E} \left[ \int_{t=s}^{s+\Delta} b(x(t), u(x(t)))  dt + \int_{t=s+\Delta}^{T} b(x(t), u(x(t)))  dt\bigg| x(s) = x \right]$$
>
> For an Itô process, the future evolution $\lbrace x(t) \rbrace_{t \geq s+\Delta}$ depends only on $x(s+\Delta)$ and is conditionally independent of the past $\{x(t)\}_{t \leq s}$ given $x(s+\Delta)$. Therefore, we can apply the tower property of conditional expectation:
>
> $$V_{f,g}(u, x, s) = \mathbb{E} \left[\int_{t=s}^{s+\Delta} b(x(t), u(x(t)))  dt\right] + \mathbb{E}\left[ \mathbb{E}\bigl[ \int_{s+\Delta}^{T} b(x(t),u(x(t))) dt \bigm| x(s+\Delta)=x' \bigr] \right],$$
> where the first expectation is taken over trajectories $x(\cdot) \sim p_{f,g}(u, x_{\text{ini}})$, and in the second term the outer expectation is with respect to $x' \sim p_{f,g}(u, x, \Delta)$ while the inner expectation is over trajectories
>  $x(\cdot) \sim p_{f,g}(u, x')$
>
> By Markov property of the Itô SDE and the definition of the value function $V_{f,g}(u, x', s+\Delta)$, we obtain:
>
> $$V_{f,g}(u, x, s) = \mathbb{E}\left[\int_{0}^{\Delta} b(x(t), u(x(t))) \, dt\right] + \mathbb{E}\left[V_{f,g}(u, x', s + \Delta)\right],$$
> where the first expectation is taken over trajectories $x(\cdot) \sim p_{f,g}(u, x)$, and the second expectation is taken over $x' \sim p_{f,g}(u, x, \Delta)$, yielding Eq. (4.1).
>
>
> **Q5.** Why the regret bound does not depend on the size of the action space?
>
> **A5.** The regret bound does not depend explicitly on the size of the action space because the analysis is driven by the statistical complexity of the density family $\{p_{f,g}(x' \mid u,x,t)\}$, rather than by a tabular enumeration of actions. In our main result, this complexity is captured by two quantities: the eluder dimension $d_{1/\varepsilon}$ of the Hellinger-distance function class $\Psi = \{\psi_{f,g}\}$ and the bracketing number $C_{1/\epsilon}$ of the conditional density class $\Upsilon = \{\upsilon_{f,g}\}$, which enter the bound only through $d_{1/\epsilon}$ and $\log C_{1/\epsilon}$ in the confidence radius $\beta$.
>
> Intuitively, both quantities measure how many “informationally distinct’’ state–action–time triples the learner must explore before the uncertainty over all models in $\mathcal F \times \mathcal G$ is resolved. The eluder dimension controls the length of the longest sequence of points at which all previously observed data are still compatible with multiple hypotheses, while the bracketing number controls how finely the conditional density family must be bracketed to approximate it in the likelihood-based analysis. Neither of these depends directly on the cardinality of the action set.
>
> **Q6.** Typos and theorem style.
>
> **A6.** We have revised according to your suggestions. Thanks!

---

> > ### Comment · Reviewer_LDX4 · 2025-11-26
> >
> > Thank you for the rebuttal. However, my biggest concern of the impossibility of implementing your algorithm still remains.
> >
> > In particular, you told me in the rebuttal that:
> > 1. "The difficulty arises only for highly expressive or infinite classes such as neural networks": I agree, but finite classes are not meaningful in the problem setting you consider with continuous state-action spaces. Optimizing over a finite set of policies and dynamics is not realistic in this setting (and the sets must also be small, otherwise you still cannot solve the optimization problem).
> > 2. "For general nonlinear function classes, including even small neural networks, finding a global maximizer is known to be NP-hard [1]": here it is not clear what you mean by "finding a global maximizer"; I assume that you mean a maximizer to the max-likelihood objective during training, since the reference you put here [1] is about this issue. While I agree with this fact, and it is a pretty obvious notion for an ML researcher nowadays to know that training a neural network involves optimizing a non-convex objective function, and so convergence to the global minima is not guaranteed (methods like ADAM can just hope to approach local minima). However, the issue in line 5 of alg 1 is definitely not about this, for two reasons: (1) first, you are *not* doing supervised learning, but you aim to maximize the expected return, which is a much more involved problem (requires RL algorithms instead of ML algorithms); (2) you are doing a max-max, once for the policies and once for the dynamics; max-max problems are well-known to be NP-HARD in general (e.g., think to bilinear problems: $\max_{x\in X,y\in\ Y}x\cdot y$ subject to convex sets $X,Y$). Clearly, (1) and (2) are very different problems from standard training of neural networks, so when you say "Our framework inherits this fundamental hardness rather than creating a new one", I completely disagree, because it is false.
> >
> > In brief, while I appreciate the theoretical nature of the paper, I am quite concerned with the lack of implementability of the proposed algorithm, which weakens the importance of the theoretical results derived (since they do not apply to the practical algorithm implemented in the appendix). For this reason, I will keep my score.

---

> > > ### Author Response · Authors · 2025-11-27
> > > **Reply to Official Comment by Reviewer LDX4**
> > >
> > > **A**: We thank the reviewer for the detailed follow-up. In light of the concerns regarding the implementability of Algorithm 1, we would like to clarify its role in our theoretical development and explain how it relates to the practical version used in the experiments.
> > >
> > > First, the purpose of Algorithm 1 is to isolate the statistical principle that yields instance-dependent performance. The max–max operator abstracts access to the optimal values of two function classes, a standard idealization in theoretical bandits and RL analysis [1–3]. In this sense, Algorithm 1 plays the same role as many idealized estimators used in RL theory. It characterizes the information-theoretic structure of the problem rather than proposing an optimization routine intended for direct implementation.
> > >
> > > Second, we fully agree that solving a general max–max problem over expressive nonlinear function classes can be NP-hard. Our analysis does not assume the ability to solve such problems in full generality. Instead, Algorithm 1 relies on an idealized oracle only to expose the instance-dependent statistical mechanism, in particular the selection of the measurable gap $\Delta_{n,k}$, on which our theoretical guarantees are built. We have highlighed it in line 206. Identifying practical surrogate optimizers for this oracle is outside the scope of our theoretical contribution and is consistent with how RL theory traditionally separates statistical analysis from computational considerations.
> > >
> > > Finally, although Algorithm 3 relaxes Algorithm 1 to achieve full implementation, it is not an arbitrary heuristic. It is intentionally designed as a practical surrogate that approximates the idealized update in Algorithm 1, retaining the maximum-likelihood estimation structure and the use of the measurable gap $\Delta_{n,k}$. Our empirical results in Appendix C.3 further support this connection. The optimal measurement gap increases monotonically with the environment volatility $\sigma$, which matches the variance-dependent scaling predicted by our theory.
> > >
> > > We hope this clarification addresses the reviewer’s concerns.
> > >
> > >
> > > [1] Abbasi-Yadkori, Y., Pál, D., & Szepesvári, C. (2011). Improved algorithms for linear stochastic bandits. Advances in neural information processing systems, 24.
> > >
> > > [2] Jin, C., Liu, Q., & Miryoosefi, S. (2021). Bellman eluder dimension: New rich classes of rl problems, and sample-efficient algorithms. Advances in neural information processing systems, 34, 13406-13418.
> > >
> > > [3] Treven, L., Hübotter, J., Sukhija, B., Dorfler, F., & Krause, A. (2023). Efficient exploration in continuous-time model-based reinforcement learning. Advances in Neural Information Processing Systems, 36, 42119-42147.

---

### Official Review · Reviewer_mhMX · 2025-11-01

**Soundness:** 3
**Presentation:** 3
**Contribution:** 3
**Rating:** 8
**Confidence:** 3

**Summary:**

This paper studies instance-dependent guarantees for continuous-time reinforcement learning (CTRL). Under some conditions, it establishes an instance-dependent second-order regret bound for CTRL. The results provides some new insights for CTRL, including robustness on choice of measurements and weaker horizon dependence compared with prior related works.

**Strengths:**

1. Interesting and novel idea: While instance-dependent analysis has been studied in standard discrete-time RL, it is interesting to see how it adapts to continuous-time RL, where irregular measurement times introduce an extra layer of complexity. This paper tackles that setting.

2. New insights for CTRL: By analyzing instance-dependent regret, the paper offers guidance for designing measurement schedules. In particular, as long as the total measurement budget is upper bounded by the total cumulative variance (the intrinsic difficulty of the problem), it is quite flexible to design measurement strategies. The regret also exhibits only logarithmic dependence on the horizon T under the assumption that cumulative reward is bounded by 1, in contrast to the exponential dependence reported in some prior work.

3. Solid theoretical development: The proofs and main results appear reasonable (although I did not check every step in detail).

4. Empirical illustration: Although this is primarily a theoretical paper, it includes numerical experiments to support the claims.

**Weaknesses:**

1. Limited intuitive exposition in the main text: The theoretical results are dense. I suggest adding a brief proof sketch in the main paper. It would also help to discuss the key challenges and obstacles specific to the continuous-time setting, and how they differ from standard discrete-time RL (e.g., [1]).

2. My biggest concern is the quadratic density assumption (Proposition 5.9). Although Appendix A.3 provides some discussion, a more general treatment would strengthen the paper. If possible, I suggest the authors to include additional examples or at least more intuition for broader density families. It seems plausible that the Eluder dimension remains tractable for richer classes, but the paper does not currently justify this.

[1] Model-based RL as a Minimalist Approach to Horizon-Free and Second-Order Bounds. Wang et. al.

**Questions:**

If model misspecification is present, how would it affect the algorithm and the theoretical guarantees? To what extent do the main insights reported here still hold (e.g., grid-shape insensitivity, variance-driven behavior)? These questions are optional.

---

> ### Author Response · Authors · 2025-11-17
> **Rebuttal to Reviewer mhMX - Part 1**
>
> **Q1.** Proof sketch in the main paper. Key challenges and obstacles specific to the continuous-time setting, and how they differ from standard discrete-time RL?
>
>
> **A1.** We appreciate the reviewer’s suggestion and have incorporated a proof sketch right after the theorem (line 390 to 414) in the revised manuscript.
>
>
> - The first challenge is the decomposition of $\text{Regret}(N)$, since the value function $V_{f,g}(u,x,t)$ is defined in continuous time and thus lacks the natural step-wise structure of discrete-time MDPs. We rely on the continuous-time one-step identity in Equation 4.1: by the Markov property, the future trajectory depends on the past only through the current state, so the distribution of $x(s+\Delta)$ is fully characterized by the transition density $p_{f,g}(u,x,\Delta)$. Applying this recursion on the measurement grid $\lbrace t_n^k \rbrace_{k=0}^{m_n}$ yields a discrete sequence of one-step relations, allowing the suboptimality gap $V_{f^\*,g^\*}(u^\*,x_{\mathrm{ini}},0) - V_{f^\*,g^\*}(u_n,x_{\mathrm{ini}},0)$
>  to be decomposed into value gaps $V_{f_n,g_n}(u_n,x_n(t_n^k),t_n^k) - V_{f_n,g_n}(u_n,x_n(t_n^{k+1}),t_n^{k+1})$ and reward-integral gaps $\mathbb{E}\left[\int_0^{\Delta_{n,k}} b(x(t),u_n(t))\,dt \right] - \int_{t_n^k}^{t_n^{k+1}} b(x_n(t),u_n(t))dt$.
> - The value gaps can be controlled using standard techniques from discrete-time analyses. The reward-integral gaps, however, are new in continuous time. Bounding the integral $\mathbb{E}\left[\int_0^{\Delta_{n,k}} b(x(t),u_n(t))\,dt \right] - \int_{t_n^k}^{t_n^{k+1}} b(x_n(t),u_n(t))\,dt$ requires knowledge of the trajectory inside each interval, which in principle demands pointwise estimation of $p_{f^\*,g^\*}$. Since pointwise convergence is unattainable under typical learning guarantees, a direct approach is infeasible. To overcome this issue, Algorithm 2 augments each interval with a single auxiliary observation sampled uniformly at time $\hat \Delta_{n,k}$. This randomization produces an unbiased Monte Carlo estimate of the reward integral and enables the construction of an additional likelihood-based confidence set that captures intra-interval behavior while keeping the measurement cost essentially unchanged.
> - The final step combines these estimates within a regret analysis that incorporates the variance term $\text{Var}^{u_n}$, which captures diffusion-driven fluctuations of the reward integral. These fluctuations accumulate at order $\Delta_{n,k}^2$, leading to the additional term $\sum_{n=1}^N \Delta_n^2$ in the final regret bound. This term is intrinsic to the continuous-time dynamics and has no analogue in the discrete-time setting.
>
>
> **Q2.** Additional examples or at least more intuition for broader density families beyond quadratic density assumption (Proposition 5.9).
>
> **A2.** Thank you for raising this concern about Proposition 5.9. In the revision we added a second example, now Proposition 5.10 (lines 370 to 377 with the proof on lines 854 to 905), where the function classes $\mathcal F$ and $\mathcal G$ have infinite cardinality and the marginal density is represented by a quadratic form in a matrix parameter. This example illustrates a richer family of continuous time dynamics while still allowing us to bound the eluder dimension and the bracketing number by quantities of order $d^{2}\log(1+B^{2}/\varepsilon^{2})$ and a polynomial in $d$ and $1/\varepsilon$. We hope this broader example and the accompanying discussion help address your concern.

---

> > ### Author Response · Authors · 2025-11-17
> > **Rebuttal to Reviewer mhMX - Part 2**
> >
> > **Q3.** How does model misspecification affect the algorithm and the theoretical guarantees?
> >
> > **A3.** Thank you for raising this question. Our algorithm and analysis can be extended to the model misspecification setting, where the true dynamics pair $(f^\ast, g^\ast)$ does not lie in $\mathcal F \times \mathcal G$. By examining the proof of Lemma B.8, the key requirement is to control the gap
> > $$\sum_{i=1}^{n-1} \sum_{k=0}^{m_i - 1}
> > \log p_{f^\ast, g^\ast}(x_i(t_i^{k+1}) \mid u_i, x_i(t_i^k), \Delta_{i,k}) -
> > \sup_{f,g \in \mathcal F \times \mathcal G}
> > \sum_{i=1}^{n-1} \sum_{k=0}^{m_i - 1}
> > \log p_{f,g}(x_i(t_i^{k+1}) \mid u_i, x_i(t_i^k), \Delta_{i,k}),$$
> > which measures how well the model class approximates the true transition density.
> >
> > To address this, we adopt a realizability-relaxation assumption: there exists an in-class pair $(f^\dagger, g^\dagger) \in \mathcal F \times \mathcal G$ such that  $p_{f^\ast, g^\ast}(x' \mid u,x,t) \le C \cdot p_{f^\dagger, g^\dagger}(x' \mid u,x,t)$  for all $x',u,x,t$. Under this assumption, the likelihood gap is at most $m_N \log C$, so the regret guarantees continue to hold after enlarging the confidence radius from $\beta$ to $\beta + m_N \log C$. The variance-driven structure of the bound remains unchanged: the leading instance-dependent term still scales with the total reward variance, while misspecification contributes only an additive approximation error. Thus, the qualitative behavior of $\text{CT-MLE}$ remains robust under moderate misspecification.
> >
> > A more realistic misspecification regime where the inequality $p_{f^\ast, g^\ast}(x' \mid u,x,t) \le C \cdot p_{f^\dagger, g^\dagger}(x' \mid u,x,t)$  does not hold pointwise presents interesting challenges and is a compelling direction for future work beyond the scope of this submission.

---

### Author Response · Authors · 2025-11-17
**To all reviewers**

Thank you for your insightful comments. Here we list our main revisions to our paper and highlight which are they for:

**1.** We add a proof sketch in the starting from line 390 to 414 in the revised paper. (**Q1** for Reviewer mhMX)

**2.** In line 337-343, We extended our setting from finite function class to infinite ones, by introducing the brackting number. We have revised our main theorem and corresponding lemmas in line 380-388 and line 1118-1123 accordingly.

In line 370-377 We have also added a new example of continuous-time dynamics that shows a low eluder dimension and low bracketing numbers. (**Q2** for Reviewer mhMX, **Q2** for Reviewer LDX4, **Q2** for Reviewer Zkqu).

**3.** In line 939-969, we have explained why the continuous-time decomsposition as shown in (4.1) holds. (**Q4** for Reviewer LDX4)

**4.** In line 2046-2054, we have added additional abalation study to study the robustness of our algorithm to the function approximator class (**Q3** for Reviewer Zkqu)

---

### Meta-Review · Area_Chair_PJA1 · 2025-12-26

**Summary:**

Reviewers broadly agreed that the paper addresses an important and timely problem in continuous-time reinforcement learning. The reviewers found the instance-dependent regret analysis novel and valuable. Some concerns were raised about assumptions used in the theory, e.g., robustness guarantee under model misspecification. The experimental resutls are limited. Algorithm 1 replies on some oracle solving max-max over policies and dynamics, which is not implementable for continous RL problems.

**Reviewer Concerns:**

Lack of Intuition and Proof Sketches in the Main Text: addressed
Model Misspecification: discussed
Implementability: not addressed
limited empirical results: Partially outstanding

**Reviewer Scores:**

Reviewer mhMX 8->8
Reviewer Zkqu 8->8
Reviewer LDX4 4->4

---

### Decision · Program_Chairs · 2026-01-26

Reject